# FSA: An Alternative Efficient Implementation of Native Sparse Attention Kernel

**Ran Yan**[1*], **Youhe Jiang**[1*], **Zhuoming Chen**[2], **Haohui Mai**[1], **Beidi Chen**[2], **Binhang Yuan**[1]

1. The Hong Kong University of Science and Technology
2. Carnegie Mellon University

`ryanaf@connect.ust.hk, youhejiang@gmail.com`
`{zhuominc, beidic}@andrew.cmu.edu, {haohui, biyuan}@ust.hk`

## ABSTRACT

Recent advances in sparse attention mechanisms have demonstrated strong potential for reducing the computational cost of long-context training and inference in large language models (LLMs). Native Sparse Attention (NSA), one state-of-the-art approach, introduces natively trainable, hardware-aligned sparse attention that delivers substantial system-level performance boosts while maintaining accuracy comparable to full attention. However, the kernel implementation of NSA forces a loop order that is only efficient with a relatively large number of query heads in each Grouped Query Attention (GQA) group, whereas existing LLMs widely adopt a much smaller number of query heads in each GQA group — such an inconsistency significantly limits the applicability of this sparse algorithmic advance. In this work, we propose **Flash Sparse Attention (FSA)**, an alternative kernel implementation that enables efficient NSA computation across a wide range of popular LLMs with a varied, smaller number of heads in each GQA group on modern GPUs. Compared to vanilla NSA kernel implementation, our empirical evaluation demonstrates that FSA achieves (i) up to $3.5\times$ and on average $1.6\times$ kernel-level latency reduction, (ii) up to $1.25\times$ and $1.09\times$ on average end-to-end training speedup on state-of-the-art LLMs, and (iii) up to $1.36\times$ and $1.11\times$ on average for prefill-phase speedup in LLM generative inference. The source code is open-sourced and publicly available at `https://github.com/Relaxed-System-Lab/Flash-Sparse-Attention`.

## 1 INTRODUCTION

Large Language Models (LLMs) with long context windows (OpenAI, 2024; Anthropic, 2024; Young et al., 2024; Dubey et al., 2024) face prohibitive computational costs due to the full attention mechanism's quadratic time and memory complexity. As sequence length increases, attention computation becomes a critical bottleneck — for instance, attention can account for 70–80% of total decoding latency at a 64k token context (Yuan et al., 2025). In extreme cases, processing a 1 million-token prompt with an 8B model can take up to 30 minutes on a single GPU (Jiang et al., 2024a). These observations underscore the urgent need for more efficient attention mechanisms in long-context LLM training and inference. A recent promising direction is to exploit sparse attention, whereby the query of each token only interacts with a subset of key and value, dramatically reducing the computation load and HBM I/O volumes. However, implementing efficient sparse attention at scale is non-trivial — In fact, the challenge of implementing high-performance kernels has become a major obstacle to deploying state-of-the-art sparse attention techniques in practice. In this paper, we want to explore: *Can we design and implement an efficient sparse attention kernel for a wide range of current LLMs to fully unleash the potential of this algorithmic advance over modern GPUs?*

Addressing the above question is crucial because adopting sparse attention in long-context LLMs could mitigate the quadratic cost and enable new applications (Xu et al., 2025a; Chen et al., 2024; Acharya et al., 2024; Wang et al., 2024). By leveraging the inherent sparsity of attention patterns, one can significantly cut down computation and memory overhead. Among such methods, one

---

[*] represents equal contribution. Correspondence to Binhang Yuan.

promising example is Natively Sparse Attention (NSA) (Yuan et al., 2025), a recently proposed sparse attention framework, which organizes keys/values into blocks and processes them via three parallel attention modules — compressed coarse-grained tokens, selected fine-grained tokens, and sliding local windows. By learning which tokens to compress or drop, NSA achieves long-context efficiency without a predefined pattern, making it a natural choice for long-context LLM training.

Nevertheless, implementing an efficient sparse attention kernel, i.e, NSA, is challenging. The core difficulty lies in implementing the sparse mechanism in NSA (i.e., computing attention score based on selectively retained fine-grained tokens), where the query of each token needs to dynamically select a different set of keys and values. Such computation results in irregular HBM access patterns on modern GPUs, where each query processes distinct selected keys/values, potentially requiring unnecessary padding for query tiles before executing warp-/warpgroup- level matrix multiply-and-accumulate instructions (e.g., `wmma` or `wgmma`), and leading to the underutilization of tensor cores.

This scattered access pattern conflicts with the GPU hardware-efficient design principle: GPUs achieve their peak mathematical throughput when the warps execute dense (no-padded) matrix multiply and accumulation instructions. Thus, current sparse attention implementations fail to translate the theoretical floating-point operations (FLOPs) reduction into wall-clock speedups.

Vanilla NSA kernel implements a two-level loop: In the outer loop, NSA kernel loads one token and batches query attention heads that share the same key and value heads; in the inner loop, NSA kernel loads selected KV block iteratively and performs attention computation. This strategy reaches kernel efficiency only when each Grouped Query Attention (GQA) (Ainslie et al., 2023) group has sufficient number of query heads, so that no-padding is required to execute PTX instructions (e.g., `wmma` or `wgmma`) on modern GPUs.[1] However, such an assumption may not hold for a wide range of popular LLMs so that the original NSA kernel efficiency could drop considerably. With an insufficient number of query heads in each GQA group, batching query heads is inefficient to satisfy this hardware requirement. Thus, the original NSA kernel implementation must pad query attention heads to meet instruction requirements, resulting in unnecessary data loading and computations.

To resolve this issue, we propose FSA, which implements optimized kernels efficient for NSA under various GQA group settings. We make the following concrete contributions:

- **Contribution 1:** We propose an alternative implementation for the NSA kernel, which exchanges the two-level loop order in NSA implementation — FSA loops over KV blocks in the outer loop and loops over query tokens in the inner loop to accelerate this system bottleneck. Since the number of query tokens that attend to a given KV block is usually much larger than the hardware required value, FSA introduces no padding, significantly reducing unnecessary kernel memory access and FLOPs, thereby facilitating faster token selection kernel execution.

- **Contribution 2:** We analyze the trade-off between vanilla NSA and FSA implementation in terms of kernel efficiency and memory accessing paradigm, which illustrates the effective design and implementation of FSA. To maximize the performance benefits of FSA kernel design, we implement dedicated optimizations for query token memory access, which is accessed in the inner loop of FSA kernel, and employ separate optimized kernels for attention result reduction.

- **Contribution 3:** We conduct empirical studies to compare FSA with vanilla NSA and full attention. Concretely, we benchmark kernel execution latencies, end-to-end training and inference prefill phase latencies for state-of-the-art LLMs. Compared to NSA, results show that FSA delivers (i) up to $3.5\times$ and on average $1.6\times$ kernel-level latency reduction, (ii) up to $1.25\times$ and $1.09\times$ on average end-to-end training speedup, and (iii) up to $1.36\times$ and $1.11\times$ on average inference prefill-phase speedup. Compared to full attention, the performance boost is further amplified.

## 2 PRELIMINARIES AND RELATED WORK

### 2.1 GPU KERNEL IMPLEMENTATION

**Parallelization in modern GPUs.** Modern GPUs utilize massive threads to execute kernels concurrently. Optimized kernel implementations typically employ two-level parallelism: (i) Thread

---

[1]Concretely, performance is downgraded due to hardware requirements on matrix shapes for warp-/warpgroup- level matrix multiply-and-accumulate instructions (e.g., `wmma` or `wgmma`) (NVIDIA, 2025), where each dimension of a matrix tile must be larger than specified value (e.g., at least 8 on Hopper GPUs).

block-level parallelism: Optimized implementations partition input matrices into multiple tiles, assign them to thread blocks, and execute computations for each thread block in parallel. Common paradigm within a single thread block follows three key steps: Load matrix tiles into the GPU's shared memory; perform computations using the loaded tiles; and store computed results to the output tensor. (ii) Warp-level parallelism: Within each thread block, optimized kernels further partition assigned matrix tiles to multiple warps — each containing 32 threads on NVIDIA GPUs (NVIDIA, 2024d) — to enable fine-grained parallel execution. Warp-level parallelism maximizes hardware efficiency through coalesced memory access and implicit synchronization within warps.

**Efficient kernel implementation.** Modern GPU architectures impose strict requirements on the shapes of matrix tiles used in low-level computations. Specifically, PTX warp-level matrix multiply-accumulate instructions (NVIDIA, 2025) require that for matrix multiplication $C = AB$, where $A \in \mathbb{R}^{m \times k}$ and $B \in \mathbb{R}^{k \times n}$, the dimensions $m$, $n$, and $k$ must satisfy minimum size requirements for single-warp processing. On NVIDIA Hopper GPUs, $m$, $n$, $k$ must be at least 8. To achieve higher efficiency, a thread block typically utilizes multiple warps for sufficient warp-level parallelism. Additionally, modern GPUs perform optimally with coalesced and contiguous data loading and storing; non-contiguous memory access leads to a lower L2 cache hit rate, thereby reducing effective memory bandwidth and degrading overall kernel efficiency.

## 2.2 ATTENTION MECHANISMS

**Full attention.** Full attention with causality (Vaswani et al., 2017; Ainslie et al., 2023)—where each query token attends to all previous KV tokens—is standard in LLM training and inference. Formally, given sequence length $N$, query/key head dimension $d_K$, value head dimension $d_V$, $h$ query heads, and $h_K$ KV heads, attention computation involves query/key/value tensor $\mathbf{Q} \in \mathbb{R}^{N \times d_K \times h}$, $\mathbf{K} \in \mathbb{R}^{N \times d_K \times h_K}$, $\mathbf{V} \in \mathbb{R}^{N \times d_V \times h_K}$. For $j$-th ($j \in \{0, 1, ..., h-1\}$) query head, $\lfloor j/h_K \rfloor$-th (ranging from 0 to $h_K$-1) key and value head, denote involved matrices as $\mathbf{Q}^j, \mathbf{K}^{\lfloor j/h_K \rfloor} \in \mathbb{R}^{N \times d_K}, \mathbf{V}^{\lfloor j/h_K \rfloor} \in \mathbb{R}^{N \times d_V}$. Full attention computation can be formalized as:

$$\mathbf{O}^j = \text{Softmax}\left(\frac{\mathbf{Q}^j(\mathbf{K}^{\lfloor j/h_K \rfloor})^T}{\sqrt{d_K}}\right) \mathbf{V}^{\lfloor j/h_K \rfloor} \tag{1}$$

On the system side, recent research (Dao, 2023; Kwon et al., 2023) has optimized full attention from various perspectives. Notably, Flash Attention (Dao, 2023) optimizes full attention with a two-level loop: Each thread block loads a block of query tokens and, while KV tokens remain, iteratively processes a block of KV tokens and accumulates intermediate results with online softmax (Milakov & Gimelshein, 2018). Results are finally written to the output tensor. This design minimizes redundant memory accesses for query and output tensors, thereby reducing attention execution latency.

**Sparse attention.** Recent efforts in sparse attention algorithms (Yuan et al., 2025; Lu et al., 2025; Lee et al., 2023; Tay et al., 2020; Zhao et al., 2019; Tang et al., 2024; Xiao et al., 2024b; Zhu et al., 2024; Lai et al., 2025; Xu et al., 2025b; Zhang et al., 2023) and system side optimizations efforts (Zhang et al., 2024b;a; 2025b) represent an emerging trend aimed at reducing attention computation costs in long-context LLM training and inference, where standard attention performs poorly due to its quadratic complexity with respect to sequence length. The most notable efforts in sparse attention include Native Sparse Attention (NSA) (Yuan et al., 2025). Formally, in NSA, for $j$-th query head, each query token $\mathbf{q}_t^j \in \mathbb{R}^{1 \times d_K}, t \in \{0, 1, ..., N-1\}$ attends to $\tilde{N} \ll N$ KV tokens via three attention mechanisms $c \in \mathcal{C}$, where $\mathcal{C} = \{\text{cmp}, \text{sel}, \text{win}\}$, representing compression, selection, and sliding window for keys and values. We denote KV tokens as $\tilde{\mathbf{K}}_c^{\lfloor j/h_K \rfloor} \in \mathbb{R}^{\tilde{N} \times d_K}, \tilde{\mathbf{V}}_c^{\lfloor j/h_K \rfloor} \in \mathbb{R}^{\tilde{N} \times d_V}$, which contains $\lfloor j/h_K \rfloor$-th KV head and a subset of KV tokens of attention mechanism $c$. Given trainable gating scores $\tau_t^c \in [0, 1]$ for three attention modules, NSA combines the three attention mechanisms as follows:

$$\mathbf{o}_t^j = \sum_{c \in \mathcal{C}} \tau_t^c \cdot \text{Softmax}\left(\frac{\mathbf{q}_t^j(\tilde{\mathbf{K}}_c^{\lfloor j/h_K \rfloor})^T}{\sqrt{d_K}}\right) \tilde{\mathbf{V}}_c^{\lfloor j/h_K \rfloor} \tag{2}$$

Notably, the NSA kernel that selectively retains fine-grained tokens is a major system bottleneck across three attention mechanisms. This point is validated in §4.4. The NSA kernel allows each

query token across query heads that share the same KV heads to attend to distinct $T$ KV blocks, each with $B_K$ contiguous KV tokens. Distinct KV block selection imposes challenges on effectively batching query tokens and performing computation with KV blocks within one thread block. Therefore, it is crucial to optimize the batching strategy for efficient NSA kernel execution.

# 3 FLASH SPARSE ATTENTION

We present FSA design and compare with vanilla NSA (§3.1), then introduce FSA implementation and optimizations (§3.2). Finally, we provide a thorough analysis of FSA performance (§3.3).

## 3.1 FSA KERNEL DESIGN

An efficient sparse attention kernel must translate theoretical FLOPs reduction into concrete savings in memory access and computation during GPU execution. Vanilla NSA kernel is insufficient in achieving this goal. As illustrated in Figure 1 (left), NSA kernel processes query tokens one by one in the outer loop and KV blocks in the inner loop, while batching query heads. However, if the number of query heads is insufficient, this method requires padding to meet the hardware's matrix multiplication shape requirements, leading to wasteful memory access and computation.

To achieve higher kernel efficiency, FSA exchanges NSA kernel loop order and processes query heads one by one, looping over KV blocks in the outer loop and batches of query tokens in the inner loop. Since the number of such tokens is typically large enough to meet hardware requirements, this strategy requires no padding and eliminates the overhead of processing padded data.

However, due to inversion of kernel loop order, FSA encounters new challenges:

- **Non-contiguous memory access for query batches.** Due to the sparse nature of NSA token selection, for one KV block, only a subset of total query tokens is involved for attention computation, and query token indices are typically non-contiguous. When processing query tokens in FSA inner loop, it is critical to minimize the negative impact of non-contiguous memory access.

- **Online softmax statistics and attention results accumulation.** Online softmax and attention results reduction for each query token across distinct KV blocks adds another layer of complexity. In the NSA token selection logic, computing the final output for a query token requires accumulating partial attention results from its distinct selected KV blocks. Since the NSA kernel's outer loop iterates over query tokens, this accumulation process can be handled within one thread block. In contrast, FSA's inverted loop order means that partial results for a single query are computed across different thread blocks, each processing a different KV block. This design necessitates a proper management strategy for accumulating attention results distributed across thread blocks.

## 3.2 FSA KERNEL IMPLEMENTATION AND OPTIMIZATION

To implement an efficient FSA kernel, we employ an optimized token selection kernel that minimizes the negative impact of non-contiguous memory access. Additionally, an online softmax and reduction kernel are designed to efficiently handle online softmax and attention result reduction.

**FSA token selection kernel.** FSA *mitigates the impact of non-contiguous memory access by employing index tensors to orchestrate data movement.* During forward pass, as illustrated in Figure 1 (right), each thread block in FSA kernel is assigned a single (Query Head, KV Block) pair. The KV block is loaded from main memory once per thread block. The kernel then iterates through batches of non-contiguous query tokens, which are loaded and stored using index tensors $\mathcal{I}_i$ and $\mathcal{O}_i$ for $i \in \{1, 2, ..., b\}$, where $b$ is the total number of KV blocks. These index tensors are pre-computed from the NSA sparse selection tensor $\mathbf{T} \in \mathbb{R}^{h_K \times N \times T}$, which records selected KV block indices for each query token. Due to the sparse nature of token selection, each KV block is attended by a subset of $N$ query tokens. Consequently, index tensor $\mathcal{I}_i$, which contains query token indices attending to current KV block, typically holds fewer than $N$ valid indices, i.e., $N_{\text{valid}} = |\mathcal{I}_i| \leq N$. To minimize the impact of non-contiguous memory access, a thread block terminates early once it has processed all valid query indices in $\mathcal{I}_i$, avoiding further memory access or computation. Concurrently, index mapping tensor $\mathcal{O}_i$ facilitates contiguous storage of intermediate results. Note that outputs from FSA token selection kernel are not final attention scores; they are partial results that are reduced

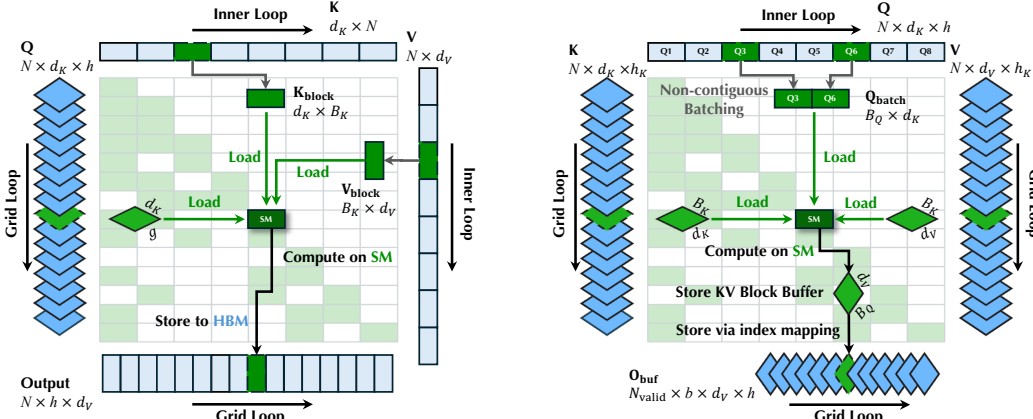

Figure 1: Left: Illustration of NSA kernel (Yuan et al., 2025), which iterates query tokens in the outer loop, and processes KV blocks in the inner loop. Right: Illustration of FSA kernel, which alternatively iterate KV blocks in the outer loop, and processes query tokens in the inner loop — partial attention results are stored in output buffer $\mathbf{O}_{\text{buf}}$ for accumulation (see §3.2 for more details).

for each query across different KV blocks in a separate reduction kernel, which we introduce next. In the backward pass, FSA kernel follows a similar logic, loading query tokens non-contiguously to compute gradients and storing intermediate gradients to buffers. The primary difference is that index tensors $\mathcal{I}_i$ and $\mathcal{O}_i$, computed during the forward pass, are retrieved from cache.

FSA *handles query attention results and gradients reduction in separate kernels.* In forward pass, FSA parallel computation of attention scores — where a single query token's results are reduced across multiple KV blocks — requires a careful implementation of online softmax and reduction logic to ensure numerical correctness. In the backward pass, a similar reduction challenge exists for gradients of query tokens. FSA achieves efficient and correct accumulation in two kernels:

**FSA reduction kernel.** Since a query's attention scores or gradients are computed across multiple thread blocks (each processing a different KV block in FSA token selection kernel), direct reduction into the output tensor in FSA kernel necessitates atomic additions (NVIDIA, 2024a) to prevent race conditions. Given the prohibitive overhead of atomic operations, FSA decouples computation from accumulation. It adopts a two-stage process:

- (i): FSA token selection kernel (see Figure 1 (right)) computes partial query attention results or gradients without reduction with online softmax and writes them to an intermediate buffer.
- (ii): A dedicated reduction kernel efficiently accumulates these partial results into a final output tensor with online softmax scaling, which we introduce next.

This two-stage arrangement effectively eliminates atomic operations and achieves efficient attention result accumulation. However, HBM memory overhead is increased due to intermediate buffers. To minimize memory overhead, we allocate a buffer sized only for $N_{\text{valid}}$ query tokens relevant to each KV block, rather than for all $N$ tokens. Index mapping tensor $\mathcal{O}_i$ facilitates contiguous I/O into this compact buffer, thereby avoiding the significant overhead of allocating a full-sized buffer for each KV block. We present a detailed analysis of FSA buffer HBM memory overhead in Appendix E.

**FSA online softmax kernel**. In the forward pass, to ensure numerical correctness, FSA needs to include online softmax statistics in two aspects:

- (i): In the FSA token selection kernel, computation results between each query token and key block must be scaled with *historical* running maximum (Milakov & Gimelshein, 2018)).
- (ii): In the reduction kernel, partial attention outputs of query tokens regarding selected KV blocks stored in the output buffer must be scaled with online softmax statistics. Additionally, final output for a query token must be scaled with log-sum exponentials (Milakov & Gimelshein, 2018).

Computing online softmax statistics within the FSA token selection kernel produces incorrect attention results. When multiple thread blocks process the same query token, each block computes only

*partial* statistics, leading to incorrect maximum values and attention outputs. To address this challenge, FSA introduces a separate online softmax kernel that pre-computes online softmax statistics using query and key tensor **Q** and key tensor **K** and stores them in a buffer.

### 3.3 FSA PERFORMANCE ANALYSIS

We analyze FSA performance by answering two critical questions regarding FSA and NSA kernel performance:

**Question 1:** *Do additional auxiliary kernels like online softmax and reduction implemented in* FSA *incur additional memory access and computation overhead?*

To answer this question, we conduct detailed memory footprint and computation load analysis and derive the following theorem:

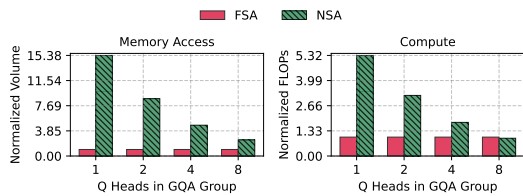

Figure 2: Comparison on memory access and FLOPs, block size is 64, top-k is 16. FSA's memory volume or FLOPs are normalized to 1.

**Theorem:** *Across popular GQA group settings, where each GQA group contains $g \in \{1, 2, 4, 8\}$ query heads, aggregate memory access volume and FLOPs of* FSA *token selection, online softmax, and reduction kernel are lower than vanilla NSA kernel.* Comparisons are presented in Figure 2. Additional memory access introduced by auxiliary kernels, i.e., FSA online softmax and reduction kernels, remains manageable, falling significantly below memory access wasted on padded data in the original NSA kernel (see more details in Appendix E).

**Question 2:** *Since* FSA *introduces non-contiguous memory access on loading query tokens and requires additional auxiliary kernels, is* FSA *generally applicable across various GPU types, and does it consistently provide performance improvements over NSA kernels?*

To answer this question, we conduct a group of micro-benchmarks and enumerate the following analysis of empirical results:

**Empirical analysis:** *Profiling results (shown in Figure 3) across various GPU types and GQA group settings confirm superior performance of* FSA. Optimized FSA outperforms vanilla NSA across popular GPU architectures and GQA group settings, despite being compro-

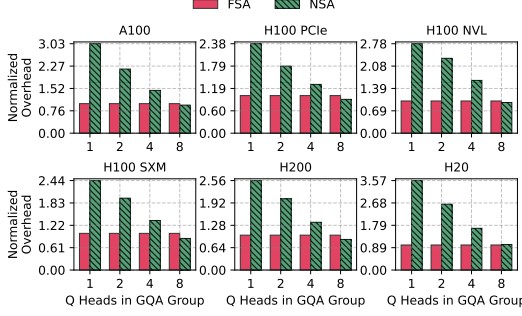

Figure 3: Real-time profiling results of the FSA and NSA kernel execution overhead across different GPUs, under block size $B_K = 64$, and top-k value $T = 16$. FSA latency is normalized to 1.

mised by non-contiguous memory access, and reducing attention results in a separate kernel. When each GQA group contains fewer than 8 query heads, FSA usually demonstrates superior performance to NSA. These empirical results demonstrate that FSA kernel's performance gains from overall reduced unnecessary memory access and FLOPs more than compensate for the overhead of non-contiguous memory access and executing multiple kernels.

## 4 EVALUATION

This section presents a comprehensive evaluation of FSA across various NSA configurations. We aim to investigate the following research questions:

- *Q1: What is the kernel-level performance of* FSA *compared with NSA and full attention across diverse NSA algorithmic configurations?*

- *Q2: What is the impact of* FSA *on end-to-end training and inference performance in practice?*

- *Q3: What is the breakdown performance of* FSA*, and how effective is each part of* FSA*?*

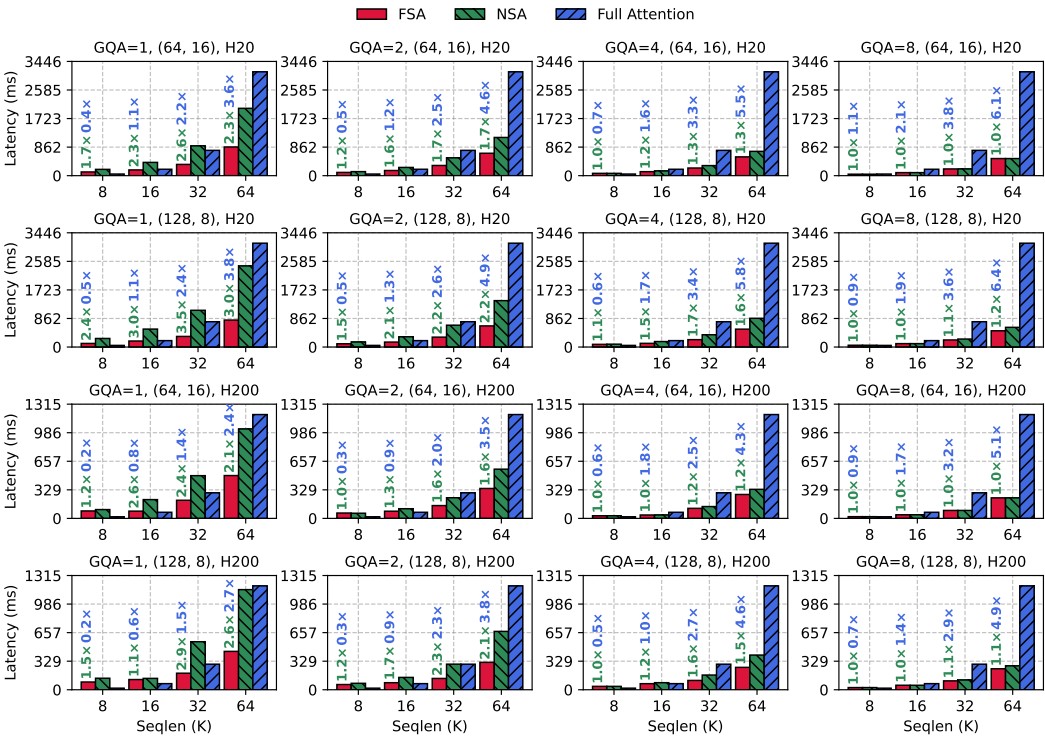

Figure 4: Performance comparison of Triton-based FSA, NSA, and full attention (enabled by Flash Attention) kernels under block sizes and top-k values of $(B_K, T)$ equals to $(64, 16)$ and $(128, 8)$.

## 4.1 EXPERIMENTAL SETUP

**Experimental setups.** We use two GPU types for evaluations: NVIDIA H20 GPUs (NVIDIA, 2024b), which provide 148 TFLOPS tensor core computational power and 4 TB/s memory bandwidth; and NVIDIA H200 GPUs (NVIDIA, 2024c), which deliver 989 TFLOPS tensor core computational power and 4.8 TB/s memory bandwidth. For end-to-end training and inference evaluations, GPUs are interconnected via NVLink, providing 450 GB/s inter-GPU bandwidth. In our evaluations, we use BF16 for training and FP16 for inference.

**Baselines.** We compare FSA with two baselines:

- **NSA (Native Sparse Attention) (Yuan et al., 2025).** Our primary baseline is vanilla NSA implementation, which introduces natively hardware-aligned trainable sparse attention. NSA maintains algorithmic performance comparable to full attention while substantially reducing computational complexity. We utilize Triton-based NSA kernel (Organization, 2024) for evaluation.

- **Full attention (Flash Attention) (Dao, 2023).** Due to limited hardware resource utilization, theoretical FLOPs reductions achieved by NSA or FSA may not translate to proportional performance gains. Therefore, the full attention baseline (with causality), which has no sparsity constraints, is essential to demonstrate the practical effectiveness of both NSA and FSA. We utilize an efficient Triton-based Flash Attention kernel(Triton, 2024) for fair comparison.

**Experimental configurations.** To ensure comprehensive evaluation, we systematically test FSA and two baselines under varying NSA configurations: (i) GQA settings $g \in \{1, 2, 4, 8\}$, where $g$ is number of query heads in one GQA group; (ii) NSA hyperparameter block size $B_K$ and top-k $T$ combinations of $(B_K, T) \in \{(64, 16), (128, 8)\}$; and (iii) sequence lengths of $\{8K, 16K, 32K, 64K\}$ tokens.[2] For end-to-end training and inference evaluations, we evaluate performance using Llama3-8B (Dubey et al., 2024), Qwen3-14B (Yang et al., 2025), and Qwen2.5-32B (Team, 2024) with

---

[2]More experiments on ultra long sequence lengths, i.e., 128K and 256K sequence lengths, are presented in Appendix F.

sequence lengths of 32K and 64K. When the entire model is too large to fit on a single GPU for training, we use pipeline parallelism (Shoeybi et al., 2019) to distribute model across multiple GPUs.

**Evaluation metrics.** Following established practices in prior research (Yuan et al., 2025; Lu et al., 2025; Dao, 2023), we employ two metrics to evaluate system efficiency: (i) Kernel execution latency, which measures computational time required for attention operations, and (ii) training and inference latency, which measures end-to-end time required to process a single batch of data during model training and inference. These metrics directly assess FSA's computational efficiency.

## 4.2 FSA KERNEL BENCHMARKING RESULTS (Q1)

**FSA kernel performance.** We evaluate the kernel performance of FSA across both H20 and H200 GPUs under various configurations. In this section, we evaluate FSA on one single GPU, while we present distributed evaluations of FSA in Appendix I. As shown in Figure 4, the evaluation results demonstrate that FSA outperforms both NSA and full attention across most of the tested scenarios:

- **Comparison with NSA.** FSA outperforms NSA with significantly lowered memory access volume and FLOPs in NSA token selection module, despite introducing non-contiguous memory access and auxiliary kernels (see details in §3). FSA achieves up to $3.5\times$ speedup and on average $1.8\times$ lower kernel latency on H20 GPUs, and up to $2.9\times$ speedup and on average $1.4\times$ lower kernel latency on H200 GPUs compared to NSA. Performance gap between FSA and NSA widens with smaller GQA group settings ($g \in \{1, 2\}$) and longer sequence lengths (32K and 64K tokens), with peak performance improvement of $3.5\times$ observed at $g = 1$ (one query head in one GQA group) and sequence length of 32K tokens. Furthermore, FSA maintains consistent performance improvements across different NSA algorithmic configurations, e.g., where $(B_K, T) = (64, 16)$ and $(B_K, T) = (128, 8)$, demonstrating robust efficiency gains across diverse parameter settings.

- **Comparison with full attention.** For long sequences, FSA outperforms full attention with an efficient NSA algorithm and even more efficient token selection. FSA achieves up to $6.4\times$ speedup and on average $2.4\times$ lower kernel latency on H20 GPUs, and up to $4.9\times$ speedup and on average $2.3\times$ lower kernel latency on H200 GPUs compared to full attention. Performance gap between FSA and full attention increases dramatically with a larger number of query heads in each GQA group, with the most substantial improvement of $6.4\times$ observed at $g = 8$ (8 query heads in one GQA group) and sequence length of 64K tokens. Similarly, FSA maintains superior efficiency across $(B_K, T) \in \{(64, 16), (128, 8)\}$ settings, demonstrating consistent and substantial performance advantages over full attention. On the other hand, vanilla NSA lags behind full attention in many tested cases, even with its sparse attention mechanism. For example, when the sequence length is 32K, one GQA group contains one query head, NSA consistently falls short of full attention, while FSA demonstrates superior performance than full attention.

## 4.3 END-TO-END PERFORMANCE COMPARISON (Q2)

**End-to-end training performance.** We benchmark end-to-end training performance of FSA against NSA and full attention across various models and hardware setups. As shown in Figure 5, results demonstrate that FSA consistently reduces training latency across all evaluated cases. Specifically, FSA achieves up to $1.25\times$ speedup and on average $1.09\times$ speedup compared to NSA, and delivers up to $2.47\times$ speedup and an average of $1.86\times$ speedup compared to full attention. These efficiency gains are pronounced with longer sequences and on higher-performance hardware like the H200, demonstrating FSA's effectiveness in accelerating computation-intensive training scenarios.

**Inference performance.** For prefill latency, we benchmark FSA against NSA and full attention across various models and hardware setups. As shown in Figure 6, our results demonstrate that FSA achieves lower prefill latency across most evaluated configurations. Specifically, FSA achieves up to $1.36\times$ speedup and on average $1.11\times$ speedup compared to NSA. FSA performance advantages are even more significant when compared to full attention, where FSA delivers up to $1.69\times$ speedup and an average of $1.39\times$ speedup. Taken together, these results underscore FSA's efficacy in accelerating the prefill phase of LLM inference.[3] In terms of decoding latency, FSA matches that of NSA, which reduces memory access of the decoding phase by only loading a sparse subset com-

---

[3]We present more detailed decoding evaluations in Appendix G.

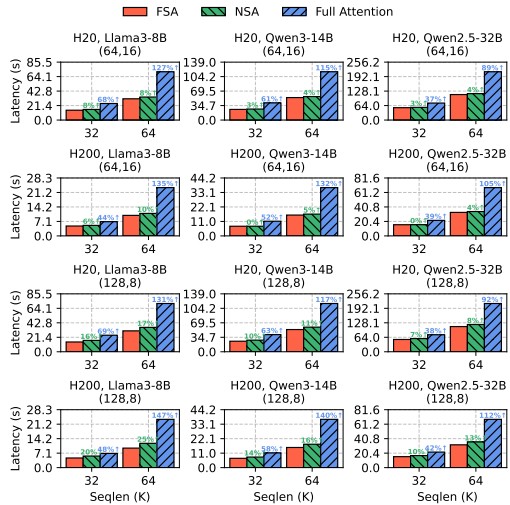

Figure 5: End-to-end training latency of FSA, NSA, full attention.

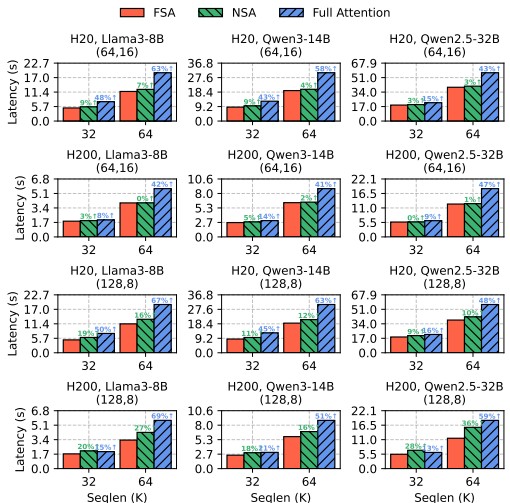

Figure 6: Inference Prefill latency of FSA, NSA, full attention.

posed of compressed tokens, selected tokens, and recent tokens from a sliding window (Yuan et al., 2025).

## 4.4 PERFORMANCE BREAKDOWN & ABLATION STUDIES (Q3)

In this section, we evaluate FSA at both kernel and end-to-end (training or inference) levels. At the kernel level, we analyze forward and backward performance separately, and examine each of the three attention mechanisms within NSA: Compression, selection, and sliding window on key/value tokens. We conduct ablation studies to assess the effectiveness of FSA kernel optimizations. We validate the implementation correctness of FSA by comparing training loss across FSA, NSA, and full attention in Appendix D.

**Forward and backward breakdown.** We conduct a detailed breakdown to analyze forward and backward attention computation latencies of FSA, NSA, and full attention across various NSA configurations. As shown in Figure 7, FSA demonstrates superior performance in both forward and backward attention computations across all evaluated scenarios. For forward computation, FSA achieves up to 2.36× speedup and on average 1.62× lower latency compared to NSA, and up to 3.23× speedup and on average 1.83× lower latency compared to full attention. Backward computation analysis reveals even more pronounced advantages, since FSA avoids computation costs for index tensors $\mathcal{I}_i$, $\mathcal{O}_i$ for $i$-th KV block (see details in §3.2). FSA achieves up to 4.32× speedup and on average 2.59× lower latency compared to NSA, and up to 7.45× speedup and on average 6.89× lower latency compared to full attention. Performance improvements remain consistent across different NSA configurations, demonstrating that FSA provides robust efficiency gains.

**Compression, selection, and sliding window breakdown.** We conduct detailed breakdown experiments for the three essential steps in NSA. As demonstrated in Figure 8, the token selection phase dominates overall attention computation performance, accounting for up to 79% and on average 65% of total attention overhead across all evaluated configurations. And FSA achieves substantial performance improvements in token selection, delivering up to 7.6× speedup and on average 3.4× lower latency compared to NSA in this critical phase. These results highlight that FSA's primary performance advantages stem from its efficient handling of token selection computation.

**Ablation study on sparse attention performance.** We present an ablation study of FSA kernel performance in Figure 9, where we disable each of additional optimizations of FSA we mentioned in §3. Results demonstrate that by disabling the inner loop (one thread block for one query batch), performance of FSA kernel drops by up to 18.9% and on average 11.9%, and by disabling early return optimization, performance drops by up to 25.2% and on average 18.2%. These empirical results demonstrate the importance of each component of our FSA optimization in enhancing performance.

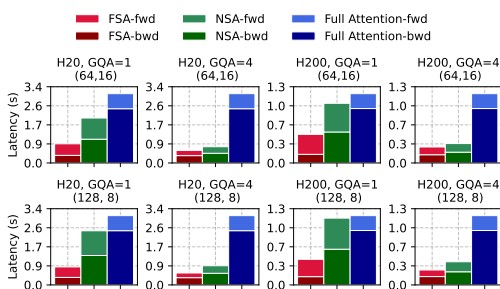

Figure 7: Experimental breakdown of FSA, NSA, and full attention latencies during forward and backward computation.

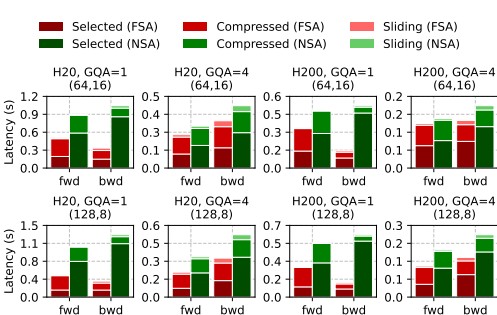

Figure 8: Experimental breakdown of token compression, selection, and sliding window attention overhead during forward/backward pass.

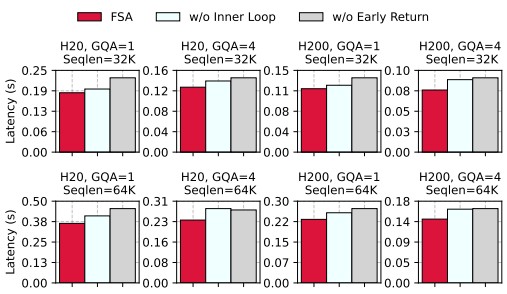

Figure 9: Ablation study (with or without FSA optimizations) on FSA kernel.

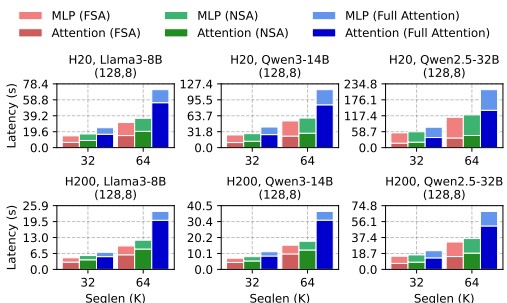

Figure 10: Breakdown of computation time for attention and MLP during end-to-end training.

**End-to-end training breakdown.** To isolate the source of performance improvements, we conduct a breakdown analysis of the end-to-end training latency. As shown in Figure 10, results demonstrate that FSA's performance improvements originate from attention computation. Within this component, FSA achieves up to $1.4\times$ and on average $1.23\times$ lower latency than NSA, and realizes a speedup of up to $3.87\times$ and on average $2.91\times$ over full attention. This analysis confirms that overall end-to-end speedup is driven by FSA's fundamental optimizations in NSA token selection.

## 5 CONCLUSION

We presented Flash Sparse Attention (FSA), a kernel design that broadens the applicability of Native Sparse Attention (NSA) to modern LLMs where each GQA group contains a small number of query heads. By inverting kernel loop order and introducing tailored optimizations for non-contiguous memory access, online softmax, and accumulation, FSA eliminates padding inefficiencies that limit NSA on current GPUs. Evaluation demonstrates that FSA achieves substantial improvements in both kernel-level and end-to-end performance, offering consistent speedups in training/inference across state-of-the-art long-context LLMs. These results highlight that algorithm–system co-design is critical for translating theoretical efficiency of sparse attention into practical acceleration. We believe FSA provides a foundation for future exploration of hardware-efficient sparse attention.

## ACKNOWLEDGMENT

This work is supported by the HKUST startup grant R9895 from CSE; RGC-ECS project 26218024; RGC-NSFC project CRS_HKUST601/24.

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

## A  THE USE OF LARGE LANGUAGE MODELS

In this paper, we leverage LLMs to enhance academic writing quality by ensuring grammatical correctness and improving sentence structure.

## B  NOTATIONS

The notations used in this paper are summarized in Table 1.

Table 1: Notations and Explanations.

| Notation | Explanation |
|---|---|
| $N$ | Sequence length. |
| $d_K$ | Head dimension for query and key tensor. |
| $d_V$ | Head dimension for value tensor. |
| $d$ | Uniform head dimension, i.e., $d = d_K = d_V$. |
| $h$ | Number of Q heads. |
| $h_K$ | Number of KV heads. |
| $g$ | GQA group size, defined as $g = \frac{h}{h_K}$. |
| $T$ | Number of selected KV blocks of each query token. (Hyperparameter of the NSA sparse attention module.) |
| $B_K$ | Block size of each KV block; a NSA hyperparameter. |
| $b$ | Number of KV blocks; $b = \frac{N}{B_K}$. |
| $B_Q$ | Query batch size in FSA; a FSA hyperparameter. |
| $\mathcal{I}_i$ | The set of query indices attending to the $i$-th KV block. ($\mathcal{I}_i$ contain non-contiguous query indices, usually $\lvert \mathcal{I}_i \rvert \leq N$.) |
| $\mathcal{O}_i$ | The output tensor mapping for the $i$-th KV block; e.g., $\mathcal{O}_i[j]$ gives the storage position of token $j$ in the output buffer. |
| $N_{\text{valid}}$ | The number of valid query tokens in $\mathcal{I}_i$. |
| $\mathbf{T}$ | Sparse selected KV block indices in NSA. |
| $\mathbf{Q},\mathbf{KV}$ | Full query, key, and value tensor for attention computation. |
| $\mathbf{Q}_{\text{batch}}$ | Non-contiguous Query batches introduced in FSA. (One thread block processes multiple $\mathbf{Q}_{\text{batch}}$.) |
| $\mathbf{K}_i,\mathbf{V}_i$ | The $i$-th KV block with $B_K$ contiguous KV tokens. |
| $\mathbf{O}_{\text{buf}}$ | Intermediate buffer which holds query attention results without scaling with online softmax in FSA. |

## C  FSA IMPLEMENTATION DETAILS

FSA is implemented using 10K lines of Python and Triton code. To optimize system performance: (i) We apply fine-grained control over FSA selected attention kernel and reduction kernel to optimize warp-level parallelism. FSA usually assigns 4 warps per thread block for FSA selected attention kernel, which contains matrix multiplication operations, to enable sufficient computational resources of a given thread block. FSA usually assigns 1 to 2 warps per thread block for reduction kernel, which mainly consists of elementwise operations. Warp assignment for reduction kernel efficiently utilizes warp-level parallelism, reducing reduction kernel execution latency. (ii) We speculatively compute online softmax statistics once per KV heads. Due to invariant nature of online softmax (Milakov & Gimelshein, 2018), correctness of FSA is maintained, while significant cost for computing online softmax statistics is amortized.

## D  FSA CORRECTNESS

**FSA correctness.** To evaluate correctness of FSA kernels, we fine-tune Llama3-8B model using ML-ArXiv-Papers dataset (Shorten, 2024). We replace attention module of Llama3-8B model with

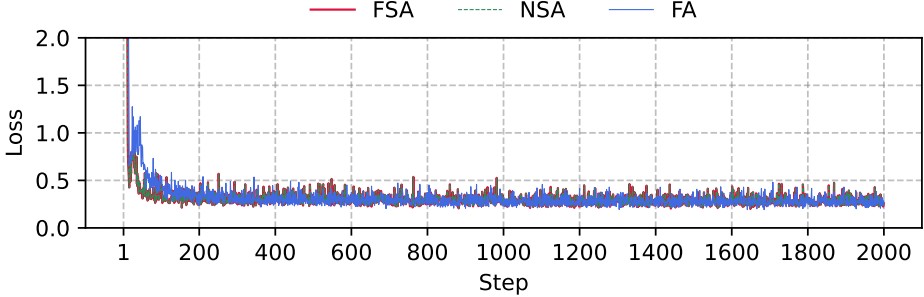

Figure 11: Loss comparison of FSA/NSA/full attention in end-to-end Llama3-8B training.

either FSA or NSA, while initializing all other components with pretrained model checkpoints provided by Meta. For fair comparison with full attention, we reinitialize the parameters of the attention module. Loss comparison among FSA, NSA, and full attention is presented in Figure 11. Results demonstrate that all three methods achieve stable and similar convergence, and FSA exhibits a similar loss curve to NSA, validating the correctness of the FSA kernel.

**Further analysis.** To strengthen our accuracy evaluation, we conduct additional experiments by fine-tuning smaller models across diverse tasks. Specifically, we fine-tune Llama-3.2-1B, Llama-3.2-3B, and Llama-3.1-8B Dubey et al. (2024) on three representative LongBench Zai (2023) tasks: multi-document QA (MQA) on the HotpotQA dataset, single-document QA (SQA) on the Qasper dataset, and synthetic-data QA (Synthetic) on the PassR-EN dataset. For Llama-3.2-1B and Llama-3.2-3B, we report the average loss after convergence over 2K training steps, comparing FSA, NSA, and Full Attention; these results appear in Table 2. To more comprehensively assess accuracy preservation, we further evaluate perplexity and QA F1 across all three models and tasks. The results, summarized in Tables 3 and 4, consistently show that FSA matches the accuracy of NSA and Full Attention.

Table 2: Converged Loss Across Datasets and Attention Modes.

| Model Size | MQA (HQA) | | | SQA (Qasper) | | | Synthetic (PassR-EN) | | |
|---|---|---|---|---|---|---|---|---|---|
| | FA | FSA | NSA | FA | FSA | NSA | FA | FSA | NSA |
| 1B | 0.200 | 0.182 | 0.187 | 0.216 | 0.191 | 0.184 | 0.231 | 0.224 | 0.231 |
| 3B | 0.173 | 0.153 | 0.166 | 0.087 | 0.082 | 0.078 | 0.123 | 0.119 | 0.118 |

Table 3: PPL Across Datasets, Models, and Attention Modes.

| Model Size | MQA (HQA) | | | SQA (Qasper) | | | Synthetic (PassR-EN) | | |
|---|---|---|---|---|---|---|---|---|---|
| | FA | FSA | NSA | FA | FSA | NSA | FA | FSA | NSA |
| 1B | 5.40 | 6.79 | 6.82 | 8.77 | 9.48 | 9.45 | 3.48 | 2.52 | 2.49 |
| 3B | 2.42 | 1.50 | 1.48 | 1.20 | 2.64 | 2.62 | 1.87 | 1.94 | 1.96 |
| 8B | 1.57 | 1.17 | 1.16 | 1.28 | 1.71 | 1.70 | 1.21 | 1.26 | 1.27 |

Table 4: QA F1 Across Datasets, Models, and Attention Modes.

| Model Size | MQA (HQA) | | | SQA (Qasper) | | | Synthetic (PassR-EN) | | |
|---|---|---|---|---|---|---|---|---|---|
| | FA | FSA | NSA | FA | FSA | NSA | FA | FSA | NSA |
| 1B | 0.05 | 0.10 | 0.11 | 0.08 | 0.07 | 0.06 | 0.22 | 0.32 | 0.31 |
| 3B | 0.28 | 0.35 | 0.33 | 0.15 | 0.11 | 0.12 | 0.39 | 0.47 | 0.48 |
| 8B | 0.32 | 0.38 | 0.37 | 0.23 | 0.20 | 0.19 | 0.83 | 0.86 | 0.86 |

# E  FSA AND NSA THEORETICAL MEMORY ACCESS AND FLOPS ANALYSIS

To demonstrate how FSA outperforms NSA selected attention, we analyze as follows. For simplicity, we assume query/key/value have the same head dimension, i.e. $d = d_K = d_V$.

**FSA analytic advantages.** *Theoretically,* FSA *introduces lower memory access volume and number of floating-point operations (FLOPs) for small GQA group sizes.* We analyze FSA/NSA as follows:

FSA *memory access volume and FLOPs.* We analyze the three key components in FSA as follows:

- **FSA selected attention kernel** launches $hb$ thread blocks, where $h$ is the number of query attention heads, and $b$ is the total number of KV blocks. For a sequence of $N$ tokens, the number of KV blocks $b = \frac{N}{B_K}$, where $B_K$ is the KV block size. In one thread block, FSA selected attention kernel runs a two-level loop. In the outer loop, it loads $2B_K d$ KV tokens; in the inner loop, it iteratively loads $B_Q d$ query tokens, performs attention computation with a FLOPs of $4B_Q B_K d$, and stores $B_Q d$ query attention results. We estimate the number of our inner loop as follows. Assume each query token attends to each KV block with equal probability. Therefore, each query token attends to a given KV block with a probability of $\frac{T}{b}$, resulting in an average number of tokens attending to a given KV block of $\frac{NT}{b}$, and an average number of query batches for one KV block of $\frac{NT}{bB_Q}$. Assuming each data occupies 2 bytes, we can calculate memory accessed in bytes by FSA selected attention kernel as $4dhN(1+T)$, and FLOPs as $4dhNB_K T$.

- **FSA online softmax kernel** operates similarly to the FSA selected attention kernel, with three key differences: It is called per KV head, omits V tensor loading and computation, and intermediate attention scores storage, storing only a single scalar value per (query token, KV block) pair. Following a similar estimation logic as FSA selected attention kernel, the online softmax kernel introduces $2dh_K N(1+T)$ memory access volume in bytes, and $2dh_K NB_K T$ FLOPs.

- **FSA reduction kernel** introduces negligible FLOPs, but for each query token, it involves loading attention results of $T$ KV blocks and storing the final attention results. Therefore, FSA reduction kernel introduces $2dhN(1+T)$ memory access in bytes.

  In total, FSA incurs $dN(6h+2h_K)(1+T)$ memory access in bytes, and $dNB_K T(4h+2h_K)$ FLOPs.

*NSA Memory access volume and FLOPs.* NSA selected attention kernel launches $h_K N$ thread blocks, where $h_K$ is the number of KV heads. In each thread block, NSA kernel runs a two-level loop. In the outer loop, NSA kernel loads one query token and $g = \frac{h}{h_K}$ Q heads that share the same KV head. Due to the hardware requirements on matrix multiplication shapes, when GQA $< 8$, NSA kernels must load 8 query heads ($8d$ elements), perform computation, and mask out the undesired computation results. In the inner loop, NSA kernel iteratively (up to $T$ times) loads one KV block ($2B_K d$ elements) and performs attention computation with a FLOPs of $32B_K d$. To maintain the causal property, i.e., avoiding query tokens to attend to future KV tokens, the actual number of KV blocks that need to be loaded and participate in computations within a thread block is on average $\frac{T}{2}$. Finally, NSA kernel stores the attention results in the output tensor, incurring $gd$ memory access. Therefore, we can estimate the memory access volume (2 bytes per data) for NSA kernel as $2dh_K N(B_K T + g + 8)$. The FLOPs for NSA kernel are $32dh_K NB_K T$.

FSA *selected attention kernels exhibit lower memory access volume and FLOPs.* With $(B_K, T) = (64, 16)$ and sequence length of 64K, which is the same configuration as presented in the NSA paper, we observe that compared to the NSA selected attention kernel, our method incurs lower memory access volume and FLOPs for GQA$\leq$8, detailed comparisons are presented in Figure 2. In particular, for GQA=4, a common configuration in LLMs, our method theoretically reduces memory access volume to 21.3% and FLOPs to 56.2% of those in NSA. Benefits from the more efficient hardware-aligned kernel design, our method substantially outperforms NSA across various GQA group sizes. Additionally, our method demonstrates superior performance as the NSA hyperparameter $B_K$ increases. This advantage stems from NSA's inherent inefficiency with larger KV blocks. Although NSA can easily skip loading KV blocks that fully violate causal property, to maintain causality constraints for KV blocks that partially violate causal property, NSA must mask out many KV tokens within the KV block, leading to wasteful memory accesses where loaded data is only partially valid for computation. As the KV block size $B_K$ grows larger, this inefficiency becomes increasingly pronounced, as a greater proportion of the loaded KV block remains unused due to causal masking. In contrast, our method processes all query tokens that attend to a given KV block within a single thread block, naturally satisfying causal constraints without requiring extensive masking. This approach achieves superior memory efficiency by ensuring that all loaded KV data contributes to the computation, resulting in significantly lower memory access overhead.

**FSA trade-offs.** FSA *trades lowered memory access volume and FLOPs with non-contiguous loading and more buffer overhead.* Theoretical advantages of FSA come at the price of involving non-contiguous memory access and more buffers that occupy HBM memory. We analyze how these factors compromise FSA performance and how FSA optimizes memory access and buffer management as follows:

- **Optimize memory access.** The non-contiguous loading on query batches, which is inefficient on modern GPUs, compromises FSA selected attention kernel performance. Modern GPUs usually operate more efficiently under coalesced and contiguous memory access, which can improve the L2-cache hit rate and thereby kernel efficiency (NVIDIA, 2024d). Therefore, the theoretical advantages of our method cannot be fully reflected in actual hardware, due to inevitably degraded performance of non-contiguous memory access. Nonetheless, to our best effort, FSA optimizes memory access with fine-grained early return mechanisms that filter out unnecessary query batches loading. For example, for $i$-th KV block, FSA compactly stores query indices in set $\mathcal{I}_i$, which is computed via a full index table. For each query token, the full index table records whether it should attend to $i$-th KV block, and $\mathcal{I}_i$ filters the tokens that do not attend to $i$-th KV block. Therefore, when all query tokens in $\mathcal{I}_i$ are exhausted, FSA returns early.

- **Optimize buffer management.** The newly introduced buffers, $\mathbf{O}_{\text{buf}}$ appeared in Figure 1 (right), bring memory overhead. FSA minimizes buffer overhead from two aspects: (i) FSA Token selection kernel processes a subset of query heads at each time, reusing the buffers for subsequent query heads computations. (ii) FSA introduces an output index mapping tensor to store results compactly. For each query head, FSA only reserves buffers for maximum query tokens that attend to a given KV block. On average, this value is $B_K T$, combining that $b = \frac{N}{B_K}$, FSA introduces an output buffer with $dNT$ elements. Assume each data in the output buffer occupies 2 bytes, for a sequence with 64K tokens, $T$ at 16, and $d$ at 128, $\mathbf{O}_{\text{buf}}$ occupies 1 GB HBM memory (This also applies for the buffer for intermediate gradients with respect to $\mathbf{Q}$). Compared to the high HBM memory capacity in modern GPUs,e.g., 96 GB HBM memory on H20 (NVIDIA, 2024b) and 141 GB memory on H200 (NVIDIA, 2024c), the additional buffer overhead in FSA remains manageable.

**Attention Sink Optimizations.** The attention sink phenomenon in NSA sparse token selection presents a challenge for FSA's buffer management strategy. The initial KV block receives attention from all query tokens, while subsequent KV blocks exhibit more selective attention patterns. This asymmetry creates a buffer allocation dilemma: In practice, FSA allocates uniform buffer sizes based on the maximum number of valid tokens across all KV blocks. However, the attention sink property forces this maximum as full sequence length, thereby negating the memory efficiency gains that FSA's sparse buffer management is designed to achieve. To address this inefficiency, we implement a dual-buffer allocation strategy. We maintain separate buffer allocations for the attention sink (first KV block) and the remaining KV blocks. The attention sink buffer accommodates the full query sequence, while buffers for subsequent KV blocks are sized according to their maximum valid query tokens, which are usually much smaller than full sequence length. This approach preserves the memory optimization benefits for the majority of KV blocks while handling the attention sink's dense connectivity requirements.

**FSA online profiling module.** *In real-world deployment,* FSA *dynamically selects kernel configuration via online profiling, and potentially falls back to original NSA implementation.* To ensure optimal performance across diverse NSA configurations, FSA incorporates a one-time online profiling mechanism. Upon its first execution with a new set of hyperparameters (e.g., sequence length, GQA group size), FSA benchmarks its kernel performance across several candidate query batch sizes (e.g., 1, 64, 128). When GQA group size is sufficiently large, a query batch size of 1 is additionally searched and serves as a potential fallback to original NSA strategy of batching query heads. Once profiling is complete, the fastest configuration is cached. All subsequent calls with the same hyperparameters directly use this optimal configuration, bypassing profiling step until hyperparameters change.

**Actual memory footprint of FSA buffers.** We conduct additional micro-benchmarks to measure the memory footprint of FSA buffers. Concretely, we set the head dimension at 128 and use the NSA hyperparameters $(B_K, T) = (64,16)$ or $(128,8)$, and report the profiled buffer overheads for sequence lengths ranging from 32K to 256K in Table 5. Under extreme cases, i.e., when the sequence length is 128K or 256K, FSA introduces 5.01GB or 12.36 GB buffer memory overhead, which is still much

smaller than the memory capacity of modern GPUs (e.g., H200 has 141GB memory). These results confirm that the FSA buffer memory overhead remains acceptable.

Table 5: Profiled Buffer Overhead.

| $(B_K, T)$ | Seqlen (K) | Profiled Buffer Overhead (GB) |
|---|---|---|
| (64, 16) | 32 | 0.52 |
| (64, 16) | 64 | 1.88 |
| (64, 16) | 128 | 5.01 |
| (64, 16) | 256 | 12.36 |
| (128, 8) | 32 | 0.26 |
| (128, 8) | 64 | 0.91 |
| (128, 8) | 128 | 2.28 |
| (128, 8) | 256 | 6.15 |

# F  EVALUATIONS FOR ULTRA LONG SEQUENCE LENGTHS.

We extend our evaluations to 128K and 256K sequence lengths. Fixing the head dimension at 128 and the number of query heads at 64, while varying the number of key and value heads, we evaluate configurations where a GQA group contains 1 to 8 query heads. Using the NSA hyperparameters with $(B_K, T)$ = (64, 16) or (128, 8), we benchmark the performance of FSA, NSA, and Full Attention (FA) on both H20 and H200 GPUs.

## F.1  INFERENCE PREFILL AND TRAINING EVALUATIONS

**Results discussion:** The experimental results in Table 6 and 7 show that FSA also outperforms NSA for ultra-long sequence lengths. For inference prefill execution latency, FSA achieves up to 1.47× speedup and an average of 1.20× lower kernel latency on H20 GPUs, and up to 1.86× speedup with an average of 1.23× lower kernel latency on H200 GPUs, compared to NSA. For training execution latency — measured over one forward and one backward pass — FSA achieves up to 1.91× speedup and an average of 1.37× lower kernel latency on H20 GPUs, and up to 2.55× speedup with an average of 1.49× lower kernel latency on H200 GPUs, relative to NSA.

Table 6: H20 GPU, Inference Prefill and Training Latency for Different $(B_K, T)$.

| $(B_K, T)$ | GQA | Seq Len (K) | FSA Fwd (s) | NSA Fwd (s) | FA Fwd (s) | FSA F + B (s) | NSA F + B (s) | FA F + B (s) |
|---|---|---|---|---|---|---|---|---|
| (64,16) | 1 | 128 | 1.42 | 2.08 | 2.64 | 2.36 | 4.51 | 12.08 |
| | 1 | 256 | 6.40 | 7.18 | 10.5 | 8.70 | 13.29 | 48.23 |
| | 2 | 128 | 0.87 | 1.17 | 2.62 | 1.79 | 2.71 | 12.04 |
| | 2 | 256 | 3.74 | 4.07 | 10.53 | 6.03 | 8.43 | 48.27 |
| | 4 | 128 | 0.52 | 0.61 | 2.65 | 1.43 | 1.75 | 12.07 |
| | 4 | 256 | 2.41 | 2.44 | 10.52 | 4.66 | 5.98 | 48.24 |
| | 8 | 128 | 0.45 | 0.45 | 2.61 | 1.38 | 1.39 | 12.05 |
| | 8 | 256 | 1.63 | 1.64 | 10.51 | 3.99 | 4.75 | 48.26 |
| (128,8) | 1 | 128 | 1.24 | 1.68 | 2.64 | 2.15 | 3.50 | 12.08 |
| | 1 | 256 | 5.34 | 7.50 | 10.5 | 7.56 | 13.65 | 48.23 |
| | 2 | 128 | 0.90 | 1.29 | 2.62 | 1.66 | 2.22 | 12.04 |
| | 2 | 256 | 3.18 | 4.24 | 10.53 | 5.40 | 8.33 | 48.27 |
| | 4 | 128 | 0.45 | 0.49 | 2.65 | 1.35 | 1.52 | 12.07 |
| | 4 | 256 | 2.10 | 2.53 | 10.52 | 4.29 | 5.55 | 48.24 |
| | 8 | 128 | 0.42 | 0.43 | 2.61 | 1.31 | 1.41 | 12.05 |
| | 8 | 256 | 1.56 | 1.68 | 10.51 | 3.74 | 4.17 | 48.28 |

Table 7: H200 GPU, Inference Prefill and Training Latency for Different $(B_K, T)$.

| $(B_K, T)$ | GQA | Seq Len (K) | FSA Fwd (s) | NSA Fwd (s) | FA Fwd (s) | FSA F + B (s) | NSA F + B (s) | FA F + B (s) |
|---|---|---|---|---|---|---|---|---|
| | 1 | 128 | 0.78 | 1.01 | 1.01 | 1.12 | 2.17 | 4.70 |
| | 1 | 256 | 3.92 | 3.97 | 3.96 | 4.81 | 6.99 | 18.75 |
| | 2 | 128 | 0.46 | 0.57 | 0.98 | 0.80 | 1.24 | 4.68 |
| | 2 | 256 | 2.14 | 2.17 | 3.98 | 3.04 | 4.12 | 18.77 |
| (64,16) | 4 | 128 | 0.27 | 0.33 | 0.99 | 0.60 | 0.82 | 4.71 |
| | 4 | 256 | 1.20 | 1.29 | 3.99 | 2.15 | 2.70 | 18.76 |
| | 8 | 128 | 0.18 | 0.18 | 0.97 | 0.50 | 0.52 | 4.67 |
| | 8 | 256 | 0.73 | 0.73 | 3.97 | 1.72 | 2.01 | 18.78 |
| | 1 | 128 | 0.65 | 1.20 | 1.00 | 0.97 | 2.47 | 4.70 |
| | 1 | 256 | 3.16 | 4.10 | 3.96 | 3.99 | 6.79 | 18.75 |
| | 2 | 128 | 0.38 | 0.66 | 0.98 | 0.70 | 1.40 | 4.68 |
| | 2 | 256 | 1.76 | 2.21 | 3.99 | 2.58 | 3.90 | 18.77 |
| (128,8) | 4 | 128 | 0.21 | 0.28 | 0.99 | 0.53 | 0.73 | 4.71 |
| | 4 | 256 | 1.06 | 1.24 | 3.97 | 1.87 | 2.44 | 18.76 |
| | 8 | 128 | 0.17 | 0.20 | 0.97 | 0.48 | 0.58 | 4.67 |
| | 8 | 256 | 0.71 | 0.75 | 3.95 | 1.52 | 1.70 | 18.78 |

## F.2 INFERENCE END-TO-END EVALUATIONS

By further fixing the number of generated tokens at 512, we evaluate the end-to-end inference execution latency of FSA, NSA, and Full Attention on both H20 and H200 GPUs.

**Results and discussion:** The experimental results in Table 8 and 9 demonstrate that FSA's performance scales well for extremely long sequences. For inference execution latency: (i) compared to NSA, FSA achieves up to $1.40\times$ speedup and on average $1.16\times$ lower kernel latency on H20 GPUs, and up to $1.59\times$ speedup and on average $1.15\times$ lower kernel latency on H200 GPUs. (ii) Compared to Full Attention, FSA achieves up to $7.20\times$ speedup and on average $4.61\times$ lower kernel latency on H20 GPUs, and up to $4.71\times$ speedup and on average $2.96\times$ lower kernel latency on H200 GPUs.

Table 8: H20 Inference Latency (s) for $(B_K, T)$ at (64,16) and (128,8).

| Method | $(B_K, T)$ | GQA = 1 | | GQA = 2 | | GQA = 4 | | GQA = 8 | |
|---|---|---|---|---|---|---|---|---|---|
| | | 128K | 256K | 128K | 256K | 128K | 256K | 128K | 256K |
| FSA | (64,16) | 1.67 | 6.71 | 1.12 | 4.05 | 0.77 | 2.72 | 0.70 | 1.94 |
| | (128,8) | 1.49 | 5.65 | 1.15 | 3.49 | 0.70 | 2.41 | 0.67 | 1.87 |
| NSA | (64,16) | 2.33 | 7.49 | 1.42 | 4.38 | 0.86 | 2.75 | 0.70 | 1.95 |
| | (128,8) | 1.93 | 7.81 | 1.54 | 4.55 | 0.74 | 2.84 | 0.68 | 1.99 |
| FA | – | 4.34 | 13.45 | 4.32 | 13.48 | 4.35 | 13.47 | 4.31 | 13.46 |

Table 9: H200 Inference Latency (s) for $(B_K, T)$ at (64,16) and (128,8).

| Method | $(B_K, T)$ | GQA = 1 | | GQA = 2 | | GQA = 4 | | GQA = 8 | |
|---|---|---|---|---|---|---|---|---|---|
| | | 128K | 256K | 128K | 256K | 128K | 256K | 128K | 256K |
| FSA | (64,16) | 1.06 | 4.24 | 0.74 | 2.46 | 0.55 | 1.52 | 0.46 | 1.05 |
| | (128,8) | 0.93 | 3.48 | 0.66 | 2.08 | 0.49 | 1.38 | 0.45 | 1.03 |
| NSA | (64,16) | 1.29 | 4.29 | 0.85 | 2.49 | 0.61 | 1.61 | 0.46 | 1.05 |
| | (128,8) | 1.48 | 4.42 | 0.94 | 2.53 | 0.56 | 1.56 | 0.48 | 1.07 |
| FA | – | 1.88 | 4.86 | 1.85 | 4.88 | 1.86 | 4.89 | 1.84 | 4.87 |

## G COMPARISON WITH FLASHDECODING

To compare with the state-of-the-art FlashDecoding kernel Dao et al. (2023), we conducted additional experiments measuring decoding execution latency for FlashDecoding and FSA. Given a

prefill sequence length (ranging from 32K to 256K), we present the average decoding latency across 1K generated tokens. We fixed the number of attention heads at 64 and the head dimension at 128. For FSA, the sparse-attention hyperparameters were set to a block size $B_K$ of 64 and TopK Value $T$ of 16.

**Result discussions:** Experimental results in Table 10 demonstrate that FSA achieves superior performance to FlashDecoding. Compared to FlashDecoding, FSA demonstrates an average speedup of 5.46x on H20 GPU and 2.16x on H200 GPU. During the decoding phase, FlashDecoding partitions the key and value tokens and distributes the resulting attention computation tasks across multiple thread blocks, thereby increasing kernel-level parallelism and improving decoding throughput. However, due to the sparsity in FSA, the FSA decoding throughput is still superior to FlashDecoding.

Table 10: Decoding Latency on H20 and H200 GPUs.

| Seq Len (K) | H20 Latency (ms) | | | H200 Latency (ms) | | |
|---|---|---|---|---|---|---|
| | FlashDecoding | NSA | FSA | FlashDecoding | NSA | FSA |
| 32 | 0.88 | 0.46 | 0.45 | 0.51 | 0.48 | 0.47 |
| 64 | 1.71 | 0.46 | 0.48 | 0.88 | 0.53 | 0.54 |
| 128 | 3.32 | 0.50 | 0.48 | 1.70 | 0.55 | 0.54 |
| 256 | 5.76 | 0.62 | 0.61 | 1.75 | 0.62 | 0.63 |

## H  COMPILIATION OVERHEAD

To determine the optimal Triton kernel hyperparameters, both FSA and NSA incur a compilation overhead. For a given NSA hyperparameter combination, this overhead occurs only once. Setting $(B_K, T)$ at (64, 16) or (128, 8), we evaluate the compilation overhead of FSA and NSA for sequence length across diverse sequence lengths. The experimental results are summarized in Table 11.

Table 11: Compilation Overhead on H20 and H200 GPUs.

| Seqlen (K) | Framework | H20 Overhead (s) | H200 Overhead (s) |
|---|---|---|---|
| 32 | FSA | 2.16 | 1.82 |
| 32 | NSA | 2.12 | 1.78 |
| 64 | FSA | 2.37 | 2.01 |
| 64 | NSA | 2.33 | 1.95 |
| 128 | FSA | 2.59 | 2.24 |
| 128 | NSA | 2.55 | 2.19 |
| 256 | FSA | 2.80 | 2.36 |
| 256 | NSA | 2.76 | 2.30 |

## I  EVALUATIONS ON DISTRIBUTED PERFORMANCE.

### I.1  DISTRIBUTED INFERENCE EVALUATION OF THE ATTENTION MODULE

Table 12: Distributed inference latency of the attention module on H20 GPU.

| $(B_K, T)$ | Seq Len (K) | Framework | TP=1 (ms) | TP=2 (ms) | TP=4 (ms) | TP=8 (ms) |
|---|---|---|---|---|---|---|
| (64, 16) | 32 | FSA | 82.50 | 45.00 | 25.94 | 16.25 |
| (64, 16) | 32 | NSA | 99.53 | 53.44 | 28.75 | 16.56 |
| (64, 16) | 64 | FSA | 195.84 | 110.63 | 61.25 | 38.63 |
| (64, 16) | 64 | NSA | 221.49 | 122.81 | 65.31 | 39.94 |
| (128, 8) | 32 | FSA | 80.31 | 43.44 | 24.69 | 15.63 |
| (128, 8) | 32 | NSA | 105.10 | 54.38 | 28.68 | 16.75 |
| (128, 8) | 64 | FSA | 187.50 | 102.68 | 56.25 | 33.75 |
| (128, 8) | 64 | NSA | 243.88 | 130.31 | 70.69 | 40.00 |

Table 13: Distributed inference latency of the attention module on H200 GPU.

| $(B_K, T)$ | Seq Len (K) | Framework | TP=1 (ms) | TP=2 (ms) | TP=4 (ms) | TP=8 (ms) |
|---|---|---|---|---|---|---|
| (64, 16) | 32 | FSA | 43.44 | 25.00 | 15.63 | 11.88 |
| (64, 16) | 32 | NSA | 50.31 | 27.19 | 17.63 | 12.69 |
| (64, 16) | 64 | FSA | 110.00 | 63.13 | 39.38 | 26.13 |
| (64, 16) | 64 | NSA | 121.56 | 66.81 | 40.75 | 27.00 |
| (128, 8) | 32 | FSA | 40.63 | 23.13 | 14.38 | 10.00 |
| (128, 8) | 32 | NSA | 59.06 | 31.25 | 17.50 | 11.63 |
| (128, 8) | 64 | FSA | 96.25 | 53.75 | 32.50 | 22.81 |
| (128, 8) | 64 | NSA | 124.38 | 65.69 | 37.50 | 25.56 |

We conduct additional experiments to evaluate the distributed inference performance of the attention module using FSA and NSA on H20 and H200 GPUs. We fix the number of query heads at 32, and the number of key and value heads at 8. This setting indicates that one GQA group contains 4 query heads. The results for both methods — measured across different NSA hyperparameters, sequence lengths, and tensor-parallel degrees — are summarized in the Table 12 and 13. Compared to NSA, FSA achieves an average speedup of 1.16x on H20 GPUs and 1.17x on H200 GPUs.

## I.2   END-TO-END DISTRIBUTED INFERENCE EVALUATION

Following the same configuration as Figure 6, we evaluate the distributed inference performance of the Llama-3-8B model on H20 and H200 GPUs. The results for NSA and FSA, measured under varying NSA hyperparameters, sequence lengths, and tensor-parallel degrees, are presented in the Table 14 and 15. Compared to NSA, FSA achieves an average speedup of 1.13x on H20 GPUs and 1.11x on H200 GPUs.

Table 14: End-to-end distributed inference latency on H20 GPU.

| $(B_K, T)$ | Seqlen (K) | Framework | TP=1 (s) | TP=2 (s) | TP=4 (s) | TP=8 (s) |
|---|---|---|---|---|---|---|
| (64, 16) | 32 | FSA | 5.28 | 2.88 | 1.66 | 1.04 |
| (64, 16) | 32 | NSA | 6.00 | 3.22 | 1.84 | 1.06 |
| (64, 16) | 64 | FSA | 11.14 | 7.08 | 3.92 | 2.60 |
| (64, 16) | 64 | NSA | 12.04 | 7.86 | 4.18 | 2.66 |
| (128, 8) | 32 | FSA | 5.14 | 2.78 | 1.58 | 1.00 |
| (128, 8) | 32 | NSA | 6.40 | 3.32 | 1.80 | 1.10 |
| (128, 8) | 64 | FSA | 12.00 | 6.38 | 3.60 | 2.16 |
| (128, 8) | 64 | NSA | 13.72 | 7.10 | 4.00 | 2.12 |

Table 15: End-to-end distributed inference latency on H200 GPU.

| $(B_K, T)$ | Seqlen (K) | Framework | TP=1 (s) | TP=2 (s) | TP=4 (s) | TP=8 (s) |
|---|---|---|---|---|---|---|
| (64, 16) | 32 | FSA | 1.95 | 1.12 | 0.70 | 0.46 |
| (64, 16) | 32 | NSA | 1.97 | 1.22 | 0.70 | 0.49 |
| (64, 16) | 64 | FSA | 4.51 | 2.83 | 1.48 | 1.09 |
| (64, 16) | 64 | NSA | 4.61 | 2.81 | 1.51 | 1.16 |
| (128, 8) | 32 | FSA | 1.82 | 1.04 | 0.64 | 0.45 |
| (128, 8) | 32 | NSA | 2.37 | 1.33 | 0.78 | 0.53 |
| (128, 8) | 64 | FSA | 3.61 | 2.13 | 1.34 | 0.98 |
| (128, 8) | 64 | NSA | 4.62 | 2.63 | 1.61 | 1.15 |

