# OpenReview forum: "FSA: An Alternative Efficient Implementation of Native Sparse Attention Kernel"
_ICLR.cc/2026/Conference — ICLR 2026 Poster_

### Official Review · Reviewer_W7t5 · 2025-10-30

**Soundness:** 3
**Presentation:** 3
**Contribution:** 3
**Rating:** 6
**Confidence:** 4

**Summary:**

The core issue addressed in this paper: NSA's cyclic ordering is only efficient when the number of query heads in the GQA group is large. However,  current attention mechanisms often employ GQA with a small number of attention heads, requiring NSA to use padding to meet hardware instruction requirements. This leads to redundant memory access and computation, preventing the theoretical reduction in FLOPs from translating into actual speed improvements.
The core solutions are loop order reversal, non-contiguous memory access optimization, staged computation and Softmax optimization, and memory-hardware co-adaptation.
Its greatest value lies in providing a hardware-friendly sparse attention solution for long-context LLM training and inference, which can be directly deployed within the existing GPU ecosystem, achieving an average 1.11x speedup in inference compared to NSA.

**Strengths:**

1. Problem Significance: Addresses the key bottleneck in deploying sparse attention: the incompatibility between native NSA kernels and mainstream LLMs' small GQA group sizes (typically 1-4 query heads).
2. Core Innovation: Inverts the NSA kernel's loop order (to "outer loop over KV blocks, inner loop over query tokens"), eliminating padding requirements for small GQA groups and significantly reducing redundant computation and memory access. Enhanced by memory management and specialized kernels.
3. Evaluation Results: Comprehensive testing shows substantial improvements: up to 3.5× lower kernel latency, 1.25× faster end-to-end training, and 1.36× faster inference prefill, with consistent performance across configurations.
4. Technical Depth: Theoretically reduces memory access and FLOPs to 21.3% and 56.2% of NSA respectively (GQA=4), complemented by practical engineering solutions for real-world deployment.

**Weaknesses:**

1. Limited Generalization to Extreme Lengths: Evaluation only up to 64K tokens, lacking analysis of ultra-long contexts (128K/256K). Attention sink effects on FSA's dual-buffer design remain unexamined at extreme scales.
2. Missing SOTA Comparisons: Only compares FSA with vanilla NSA and full attention. Omits comparisons with recent sparse kernels like flashdecoding, limiting perspective on performance trade-offs.
3. Insufficient Accuracy Analysis: Accuracy claims rely solely on Llama3-8B results. Lacks validation across diverse tasks and smaller models, with no study of numerical stability impact from optimizations.
4. Unquantified Deployment Overhead: Intermediate buffer costs are mentioned but not measured for memory-constrained scenarios. No discussion of integration complexity with mainstream frameworks.

**Questions:**

1. Can FSA adapt to other sparse patterns (e.g., block-sparse, dynamic)? Is the loop inversion strategy compatible beyond NSA's structure?
2.  Any evaluation beyond 64K length? How does attention sink affect the dual-buffer design?
3. Beyond loss curves, can you provide multi-model metrics (e.g., perplexity, QA F1) across diverse tasks?
4. What's the actual memory footprint of intermediate buffers? Any compilation time or distributed inference benchmarks?
5. What about other related work,like flashdecoding?

---

> ### Author Response · Authors · 2025-11-20
>
> > W1.1 & Q2.1: Limited Generalization to Extreme Lengths: Evaluation only up to 64K tokens, lacking analysis of ultra-long contexts (128K/256K). Any evaluation beyond 64K length?
> >
>
> We extend our evaluations to 128K and 256K sequence lengths. Fixing the head dimension at 128 and the number of query heads at 64, while varying the number of key and value heads, we evaluate configurations where a GQA group contains 1 to 8 query heads. Using the NSA hyperparameters with (Block Size, TopK) = (64, 16) or (128, 8), we benchmark the performance of FSA, NSA, and Full Attention (FA) on both H20 and H200 GPUs. The corresponding results are summarized in the tables below and have been incorporated into Appendix F of our revised paper.
>
> **Results discussion:** The experimental results show that FSA also outperforms NSA for ultra-long sequence lengths. For inference prefill execution latency, FSA achieves up to 1.47$\times$ speedup and an average of 1.20$\times$ lower kernel latency on H20 GPUs, and up to 1.86$\times$ speedup with an average of 1.23$\times$ lower kernel latency on H200 GPUs, compared to NSA. For training execution latency — measured over one forward and one backward pass — FSA achieves up to 1.91$\times$ speedup and an average of 1.37$\times$ lower kernel latency on H20 GPUs, and up to 2.55$\times$ speedup with an average of 1.49$\times$ lower kernel latency on H200 GPUs, relative to NSA.
>
> **Tables of experimental results:**
>
> - H20, Block Size = 64, TopK = 16
>
> | GQA | Seq Len (K) | FSA Fwd E2E (s) | NSA Fwd E2E (s) | FA Fwd E2E (s) | FSA Fwd + Bwd (s) | NSA Fwd + Bwd (s) | FA Fwd + Bwd (s) |
> | --- | --- | --- | --- | --- | --- | --- | --- |
> | 1 | 128 | 1.42  | 2.08  | 2.64 | 2.36  | 4.51  | 12.08 |
> | 1 | 256 | 6.40  | 7.18  | 10.5 | 8.70  | 13.29  | 48.23 |
> | 2 | 128 | 0.87  | 1.17  | 2.62 | 1.79  | 2.71  | 12.04 |
> | 2 | 256 | 3.74  | 4.07  | 10.53 | 6.03  | 8.43  | 48.27 |
> | 4 | 128 | 0.52  | 0.61  | 2.65 | 1.43  | 1.75  | 12.07 |
> | 4 | 256 | 2.41  | 2.44  | 10.52 | 4.66  | 5.98  | 48.24 |
> | 8 | 128 | 0.45  | 0.45  | 2.61 | 1.38  | 1.39  | 12.05 |
> | 8 | 256 | 1.63  | 1.64  | 10.51 | 3.99  | 4.75  | 48.26 |
> - H20, Block Size = 128, TopK = 8
>
> | GQA | Seq Len (K) | FSA Fwd E2E (s) | NSA Fwd E2E (s) | FA Fwd E2E (s) | FSA Fwd + Bwd (s) | NSA Fwd + Bwd (s) | FA Fwd + Bwd (s) |
> | --- | --- | --- | --- | --- | --- | --- | --- |
> | 1 | 128 | 1.24  | 1.68  | 2.64 | 2.15  | 3.50  | 12.08 |
> | 1 | 256 | 5.34  | 7.50  | 10.5 | 7.56  | 13.65  | 48.23 |
> | 2 | 128 | 0.90  | 1.29  | 2.62 | 1.66  | 2.22  | 12.04 |
> | 2 | 256 | 3.18  | 4.24  | 10.53 | 5.40  | 8.33  | 48.27 |
> | 4 | 128 | 0.45  | 0.49  | 2.65 | 1.35  | 1.52  | 12.07 |
> | 4 | 256 | 2.10  | 2.53  | 10.52 | 4.29  | 5.55  | 48.24 |
> | 8 | 128 | 0.42  | 0.43  | 2.61 | 1.31  | 1.41  | 12.05 |
> | 8 | 256 | 1.56  | 1.68  | 10.51 | 3.74  | 4.17  | 48.28 |
> - H200, Block Size = 64, TopK = 16
>
> | GQA | Seq Len (K) | FSA Fwd E2E (s) | NSA Fwd E2E (s) | FA Fwd E2E (s) | FSA Fwd + Bwd (s) | NSA Fwd + Bwd (s) | FA Fwd + Bwd (s) |
> | --- | --- | --- | --- | --- | --- | --- | --- |
> | 1 | 128 | 0.78  | 1.01  | 1.01 | 1.12  | 2.17  | 4.70 |
> | 1 | 256 | 3.92  | 3.97  | 3.96 | 4.81  | 6.99  | 18.75 |
> | 2 | 128 | 0.46  | 0.57  | 0.98 | 0.80  | 1.24  | 4.68 |
> | 2 | 256 | 2.14  | 2.17  | 3.98 | 3.04  | 4.12  | 18.77 |
> | 4 | 128 | 0.27  | 0.33  | 0.99 | 0.60  | 0.82  | 4.71 |
> | 4 | 256 | 1.20  | 1.29  | 3.99 | 2.15  | 2.70  | 18.76 |
> | 8 | 128 | 0.18  | 0.18  | 0.97 | 0.50  | 0.52  | 4.67 |
> | 8 | 256 | 0.73  | 0.73  | 3.97 | 1.72  | 2.01  | 18.78 |
> - H200, Block Size = 128, TopK = 8
>
> | GQA | Seq Len (K) | FSA Fwd E2E (s) | NSA Fwd E2E (s) | FA Fwd E2E (s) | FSA Fwd + Bwd (s) | NSA Fwd + Bwd (s) | FA Fwd + Bwd (s) |
> | --- | --- | --- | --- | --- | --- | --- | --- |
> | 1 | 128 | 0.65  | 1.20  | 1.00  | 0.97  | 2.47  | 4.70 |
> | 1 | 256 | 3.16  | 4.10  | 3.96  | 3.99  | 6.79  | 18.75 |
> | 2 | 128 | 0.38  | 0.66  | 0.98  | 0.70  | 1.40  | 4.68 |
> | 2 | 256 | 1.76  | 2.21  | 3.99  | 2.58  | 3.90  | 18.77 |
> | 4 | 128 | 0.21  | 0.28  | 0.99  | 0.53  | 0.73  | 4.71 |
> | 4 | 256 | 1.06  | 1.24  | 3.97  | 1.87  | 2.44  | 18.76 |
> | 8 | 128 | 0.17  | 0.20  | 0.97  | 0.48  | 0.58  | 4.67 |
> | 8 | 256 | 0.71  | 0.75  | 3.95  | 1.52  | 1.70  | 18.78 |

---

> > ### Author Response · Authors · 2025-11-20
> >
> > > W1.2 & Q2.2:  Attention sink effects on FSA's dual-buffer design remain unexamined at extreme scales. How does attention sink affect the dual-buffer design?
> > >
> >
> > **Attention sink effects on FSA’s dual buffer design.** The attention sink mechanism only influences the FSA buffer size for the very first KV block (i.e., only one of the dual buffers). Because the first KV block is always selected by all query tokens, FSA must allocate a buffer of size N $\times$ d to store the intermediate results, where N is the sequence length and d is the head dimension.
> >
> > **Micro-benchmark of attention sink effects.** We conduct additional micro-benchmarks to evaluate the effect of attention sink under extremely long contexts. Setting the head dimension at 128 and using BFloat16 for the buffer, we summarize the resulting buffer HBM memory consumption for various sequence lengths in the following table.
> >
> > | Seq Len (K) | HBM Buffer Overhead for the First KV Block (MB) |
> > | --- | --- |
> > | 32 | 7.99 |
> > | 64 | 15.98 |
> > | 128 | 31.96 |
> > | 256 | 63.93 |
> >
> > From the table, we observe that even at extreme scales, the attention sink introduces only minor overhead to the FSA dual-buffer design. For example, at a sequence length of 256K, the buffer overhead attributable to the attention sink is merely 63.93 MB — representing only 0.04% of the total HBM capacity of the H200 GPU.
> >
> > > W2 & Q5: Missing SOTA Comparisons: Only compares FSA with vanilla NSA and full attention. Omits comparisons with recent sparse kernels like flashdecoding, limiting perspective on performance trade-offs. What about other related work, like FlashDecoding?
> > >
> >
> > We appreciate this suggestion — to compare with the state-of-the-art FlashDecoding kernel, we conducted additional experiments measuring decoding execution latency for FlashDecoding and FSA. Given a prefill sequence length (ranging from 32K to 256K), we present the average decoding latency across 1K generated tokens. We fixed the number of attention heads at 64 and the head dimension at 128. For FSA, the sparse-attention hyperparameters were set to a block size of 64 and TopK of 16.
> >
> > **Result discussions:** Experimental results demonstrate that FSA achieves superior performance to FlashDecoding. Compared to FlashDecoding, FSA demonstrates an average speedup of 5.46x on H20 GPU and 2.16x on H200 GPU. During the decoding phase, FlashDecoding partitions the key and value tokens and distributes the resulting attention computation tasks across multiple thread blocks, thereby increasing kernel-level parallelism and improving decoding throughput. However, due to the sparsity in FSA, the FSA decoding throughput is still superior to FlashDecoding.
> >
> > **Tables of experimental results:**
> >
> > - H20:
> >
> > | Seq Len (K) | FlashDecoding Latency (ms) | FSA Decode Latency (ms) |
> > | --- | --- | --- |
> > | 32 | 0.88  | 0.45  |
> > | 64 | 1.71  | 0.48  |
> > | 128 | 3.32  | 0.48  |
> > | 256 | 5.76  | 0.61  |
> > - H200:
> >
> > | Seq Len (K) | FlashDecoding Latency (ms) | FSA Decode Latency (ms) |
> > | --- | --- | --- |
> > | 32 | 0.51  | 0.47  |
> > | 64 | 0.88  | 0.54  |
> > | 128 | 1.70  | 0.54  |
> > | 256 | 1.75  | 0.63  |
> >
> > We have integrated these evaluations in Appendix G of our updated paper.

---

> > > ### Author Response · Authors · 2025-11-20
> > >
> > > > W3: Insufficient Accuracy Analysis: Accuracy claims rely solely on Llama3-8B results. Lacks validation across diverse tasks and smaller models, with no study of numerical stability impact from optimizations.
> > > >
> > >
> > > To further enhance our accuracy analysis, we conduct additional experiments by fine-tuning smaller models across diverse tasks. In particular, we report the **average loss after convergence** for fine-tuning Llama-3.2-1B and Llama-3.2-3B [1] models. The selected tasks include single-document QA (SQA), multi-document QA (MQA), and synthetic-data QA (Synthetic). We use datasets from LongBench [2], including Qasper for SQA, HotpotQA (HQA) for MQA, and Passage-Retrieval-EN (PassR-EN) for the Synthetic task.
> > >
> > > We present the average loss after convergence across 2K training steps for all attention mechanisms. The results are summarized in the following table. These additional experimental results are included in Appendix D of our updated paper.
> > >
> > > **Table of experimental results:**
> > >
> > > | Model Size | Dataset | Attention Mode | Converged Loss  |
> > > | --- | --- | --- | --- |
> > > | 1B | HQA | FA | 0.200 ± 0.040 |
> > > | 1B | HQA | FSA | 0.182 ± 0.034 |
> > > | 1B | HQA | NSA | 0.187 ± 0.033 |
> > > | 1B | Qasper | FA | 0.216 ± 0.038 |
> > > | 1B | Qasper | FSA | 0.191 ± 0.029 |
> > > | 1B | Qasper | NSA | 0.184 ± 0.027 |
> > > | 1B | PassR-EN | FA | 0.231 ± 0.042 |
> > > | 1B | PassR-EN | FSA | 0.224 ± 0.036 |
> > > | 1B | PassR-EN | NSA | 0.231 ± 0.038 |
> > > | 3B | HQA | FA | 0.173 ± 0.036 |
> > > | 3B | HQA | FSA | 0.153 ± 0.032 |
> > > | 3B | HQA | NSA | 0.166 ± 0.034 |
> > > | 3B | Qasper | FA | 0.087 ± 0.033 |
> > > | 3B | Qasper | FSA | 0.082 ± 0.023 |
> > > | 3B | Qasper | NSA | 0.078 ± 0.022 |
> > > | 3B | PassR-EN | FA | 0.123 ± 0.035 |
> > > | 3B | PassR-EN | FSA | 0.119 ± 0.022 |
> > > | 3B | PassR-EN | NSA | 0.118 ± 0.024 |
> > >
> > > Reference:
> > >
> > > [1] Dubey, Abhimanyu, et al. "The llama 3 herd of models." *arXiv e-prints* (2024).
> > >
> > > [2] Zai, Organization. "LongBench: A Benchmark for Long-Context Language Models." https://huggingface.co/datasets/zai-org/LongBench (2023).
> > >
> > > > W4.1 & Q4.1: Unquantified Deployment Overhead: Intermediate buffer costs are mentioned but not measured for memory-constrained scenarios. What's the actual memory footprint of intermediate buffers?
> > > >
> > >
> > > **Actual memory footprint of FSA buffers.** We conduct additional micro-benchmarks to measure the memory footprint of FSA buffers. Concretely, we set the head dimension at 128 and use the NSA hyperparameters (Block Size, TopK)=(64,16) or (128,8), and report the profiled buffer overheads for sequence lengths ranging from 32K to 256K.  Under extreme cases, i.e., when the sequence length is 128K or 256K, FSA introduces 5.01GB or 12.36 GB buffer memory overhead, which is still much smaller than the memory capacity of modern GPUs (e.g., H200 has 141GB memory). These results confirm that the FSA buffer memory overhead remains acceptable.
> > >
> > > **Table of experimental results:**
> > >
> > > | (Block Size, TopK) | Seq Len (K) | Profiled Buffer Overhead (GB) |
> > > | --- | --- | --- |
> > > | (64, 16) | 32 | 0.52 |
> > > | (64, 16) | 64 | 1.88 |
> > > | (64, 16) | 128 | 5.01 |
> > > | (64, 16) | 256 | 12.36 |
> > > | (128, 8) | 32 | 0.26 |
> > > | (128, 8) | 64 | 0.91 |
> > > | (128, 8) | 128 | 2.28 |
> > > | (128, 8) | 256 | 6.15 |
> > >
> > > We have integrated these evaluations in Appendix E of our updated paper.
> > >
> > > > W4.2: No discussion of integration complexity with mainstream frameworks.
> > > >
> > >
> > > FSA can be seamlessly integrated into mainstream frameworks with minimal engineering effort. Like Full Attention, FSA exposes an interface that accepts input data and sparse attention hyperparameters and returns the resulting attention output. For instance, within the *transformers* library, FSA can be incorporated into Llama models simply by replacing the `LlamaAttention` module with the FSA module. For LLM inference serving systems, e.g., SGLang and vLLM, we could also integrate FSA under the attention kernel implementations.
> > >
> > > > Q1: Can FSA adapt to other sparse patterns (e.g., block-sparse, dynamic)? Is the loop inversion strategy compatible beyond NSA's structure?
> > > >
> > >
> > > FSA can accommodate any **dynamic sparse patterns** where each query token attends to a distinct set of KV blocks. In such cases, the **loop-inversion strategy** remains compatible as long as the dynamic sparse pattern supplies the required token-selection indices that specify the KV blocks attended by each query token. On the other hand, in **block-sparse attention patterns**, where a block of query tokens attends to the same set of KV blocks, FSA’s loop-inversion strategy is no longer applicable. In such cases, the padding inefficiencies on query tokens no longer exist.
> > >
> > > The loop inversion strategy is compatible with structures beyond NSA. For example, it can also be applied to MoBA’s [1] sparse attention design.
> > >
> > > Reference:
> > >
> > > [1] Lu, Enzhe, et al. "MoBA: Mixture of Block Attention for Long-Context LLMs." *Advances in neural information processing systems* (2025).

---

> > > > ### Author Response · Authors · 2025-11-20
> > > >
> > > > > Q3: Beyond loss curves, can you provide multi-model metrics (e.g., perplexity, QA F1) across diverse tasks?
> > > > >
> > > >
> > > > **Additional accuracy evaluations across diverse tasks with multi-model metrics:** To further validate that FSA preserves accuracy relative to NSA, we conducted additional experiments fine-tuning Llama-3.2-1B, Llama-3.2-3B, and Llama-3.1-8B [1] across diverse tasks. The selected tasks include single-document QA (SQA), multi-document QA (MQA), and synthetic-data QA (Synthetic). We use datasets from LongBench [1]: Qasper for SQA, HotpotQA (HQA) for MQA, and Passage-Retrieval-EN (PassR-EN) for the Synthetic task. We compare the perplexity and QA F1 across FSA, NSA, and Full Attention (FA). The experimental results are summarized in the following tables.
> > > >
> > > > **Tables of experimental results:**
> > > >
> > > > - Llama-3.2-1B:
> > > >
> > > > | Task Type | Dataset | Attention Mode | PPL | QA F1 |
> > > > | --- | --- | --- | --- | --- |
> > > > | SQA | Qasper | FSA | 9.48 | 0.07 |
> > > > | SQA | Qasper | NSA | 9.45 | 0.06 |
> > > > | SQA | Qasper | FA | 8.77  | 0.08  |
> > > > | MQA | HQA | FSA | 6.79 | 0.10 |
> > > > | MQA | HQA | NSA | 6.82  | 0.11 |
> > > > | MQA | HQA | FA | 5.40  | 0.05 |
> > > > | Synthetic | PassR-EN | FSA | 2.52 | 0.32 |
> > > > | Synthetic | PassR-EN | NSA | 2.49 | 0.31 |
> > > > | Synthetic | PassR-EN | FA | 3.48  | 0.22  |
> > > > - Llama-3.2-3B:
> > > >
> > > > | Task Type | Dataset | Attention Mode | PPL | QA F1 |
> > > > | --- | --- | --- | --- | --- |
> > > > | SQA | Qasper | FSA | 2.64 | 0.11 |
> > > > | SQA | Qasper | NSA | 2.62 | 0.12 |
> > > > | SQA | Qasper | FA | 1.20  | 0.15  |
> > > > | MQA | HQA | FSA | 1.50 | 0.35 |
> > > > | MQA | HQA | NSA | 1.48 | 0.33 |
> > > > | MQA | HQA | FA | 2.42  | 0.28  |
> > > > | Synthetic | PassR-EN | FSA | 1.94 | 0.47 |
> > > > | Synthetic | PassR-EN | NSA | 1.96 | 0.48 |
> > > > | Synthetic | PassR-EN | FA | 1.87  | 0.39  |
> > > > - Llama-3.1-8B:
> > > >
> > > > | Task Type | Dataset | Attention Mode | PPL | QA F1 |
> > > > | --- | --- | --- | --- | --- |
> > > > | SQA | Qasper | FSA | 1.71 | 0.20 |
> > > > | SQA | Qasper | NSA | 1.7 | 0.19 |
> > > > | SQA | Qasper | FA | 1.28 | 0.23 |
> > > > | MQA | HQA | FSA | 1.17 | 0.38 |
> > > > | MQA | HQA | NSA | 1.16 | 0.37 |
> > > > | MQA | HQA | FA | 1.57 | 0.32 |
> > > > | Synthetic | PassR-EN | FSA | 1.26 | 0.86 |
> > > > | Synthetic | PassR-EN | NSA | 1.27 | 0.86 |
> > > > | Synthetic | PassR-EN | FA | 1.21 | 0.83 |
> > > >
> > > > These additional experimental results are included in Appendix D of our updated paper.
> > > >
> > > > Reference:
> > > >
> > > > [1] Dubey, Abhimanyu, et al. "The llama 3 herd of models." *arXiv e-prints* (2024).
> > > >
> > > > [2] Zai, Organization. "LongBench: A Benchmark for Long-Context Language Models." https://huggingface.co/datasets/zai-org/LongBench (2023).

---

> > > > > ### Author Response · Authors · 2025-11-20
> > > > >
> > > > > > Q4.2: Any compilation time or distributed inference benchmarks?
> > > > > >
> > > > >
> > > > > **Compilation time evaluation:** To determine the optimal Triton kernel hyperparameters, both FSA and NSA incur a compilation overhead. For a given NSA hyperparameter combination, this overhead occurs only once. Setting (Block, TopK) at (64, 16) or (128, 8), we evaluate the compilation overhead of FSA and NSA for sequence length across diverse sequence lengths. The experimental results are summarized in the following table.
> > > > >
> > > > > | Seq Len (K) | Framework | H20 Compilation Overhead (s) | H200 Compilation Overhead (s) |
> > > > > | --- | --- | --- | --- |
> > > > > | 32 | FSA | 2.16 | 1.82 |
> > > > > | 32 | NSA | 2.12 | 1.78 |
> > > > > | 64 | FSA | 2.37 | 2.01 |
> > > > > | 64 | NSA | 2.33 | 1.95 |
> > > > > | 128 | FSA | 2.59 | 2.24 |
> > > > > | 128 | NSA | 2.55 | 2.19 |
> > > > > | 256 | FSA | 2.80 | 2.36 |
> > > > > | 256 | NSA | 2.76 | 2.30 |
> > > > >
> > > > > **Distributed Inference Evaluation of the Attention Module:** We conduct additional experiments to evaluate the distributed inference performance of the attention module using FSA and NSA on H20 and H200 GPUs. We fix the number of query heads at 32, and the number of key and value heads at 8. This setting indicates that one GQA group contains 4 query heads. The results for both methods — measured across different NSA hyperparameters, sequence lengths, and tensor-parallel degrees — are summarized in the tables below. Compared to NSA, FSA achieves an average speedup of 1.16$\times$ on H20 GPUs and 1.17$\times$ on H200 GPUs.
> > > > >
> > > > > - H20 GPU:
> > > > >
> > > > > | (Block Size, TopK) | Seq Len (K) | Framework | TP=1 (ms) | TP=2 (ms) | TP=4 (ms) | TP=8 (ms) |
> > > > > | --- | --- | --- | --- | --- | --- | --- |
> > > > > | (64, 16) | 32 | FSA | 82.50  | 45.00  | 25.94  | 16.25  |
> > > > > | (64, 16) | 32 | NSA | 99.53 | 53.44  | 28.75  | 16.56  |
> > > > > | (64, 16) | 64 | FSA | 195.84  | 110.63  | 61.25  | 38.63  |
> > > > > | (64, 16) | 64 | NSA | 221.49  | 122.81  | 65.31  | 39.94  |
> > > > > | (128, 8) | 32 | FSA | 80.31  | 43.44  | 24.69  | 15.63  |
> > > > > | (128, 8) | 32 | NSA | 105.10  | 54.38  | 28.68  | 16.75  |
> > > > > | (128, 8) | 64 | FSA | 187.50  | 102.68  | 56.25  | 33.75  |
> > > > > | (128, 8) | 64 | NSA | 243.88  | 130.31  | 70.69  | 40.00  |
> > > > > - H200 GPU:
> > > > >
> > > > > | (Block Size, TopK) | Seq Len (K) | Framework | TP=1 (ms) | TP=2 (ms) | TP=4 (ms) | TP=8 (ms) |
> > > > > | --- | --- | --- | --- | --- | --- | --- |
> > > > > | (64, 16) | 32 | FSA | 43.44  | 25.00  | 15.63  | 11.88  |
> > > > > | (64, 16) | 32 | NSA | 50.31  | 27.19  | 17.63  | 12.69  |
> > > > > | (64, 16) | 64 | FSA | 110.00  | 63.13  | 39.38  | 26.13  |
> > > > > | (64, 16) | 64 | NSA | 121.56  | 66.81  | 40.75  | 27.00  |
> > > > > | (128, 8) | 32 | FSA | 40.63  | 23.13  | 14.38  | 10.00  |
> > > > > | (128, 8) | 32 | NSA | 59.06  | 31.25  | 17.50  | 11.63  |
> > > > > | (128, 8) | 64 | FSA | 96.25  | 53.75  | 32.50  | 22.81  |
> > > > > | (128, 8) | 64 | NSA | 124.38  | 65.69  | 37.50  | 25.56  |
> > > > >
> > > > > **End-to-end distributed inference evaluation:** Following the same configuration as Figure 6 of our paper, we evaluate the distributed inference performance of the Llama-3-8B model on H20 and H200 GPUs. The results for NSA and FSA, measured under varying NSA hyperparameters, sequence lengths, and tensor-parallel degrees, are presented in the tables below. Compared to NSA, FSA achieves an average speedup of 1.13$\times$ on H20 GPUs and 1.11$\times$ on H200 GPUs.
> > > > >
> > > > > - On H20 GPUs:
> > > > >
> > > > > | (Block Size, TopK) | Seq Len (K) | Framework | TP=1 (s) | TP=2 (s) | TP=4 (s) | TP=8 (s) |
> > > > > | --- | --- | --- | --- | --- | --- | --- |
> > > > > | (64, 16) | 32 | FSA | 5.28  | 2.88  | 1.66  | 1.04  |
> > > > > | (64, 16) | 32 | NSA | 6.00  | 3.22  | 1.84  | 1.06  |
> > > > > | (64, 16) | 64 | FSA | 11.14  | 7.08  | 3.92  | 2.60  |
> > > > > | (64, 16) | 64 | NSA | 12.04  | 7.86  | 4.18  | 2.66  |
> > > > > | (128, 8) | 32 | FSA | 5.14  | 2.78  | 1.58  | 1.00  |
> > > > > | (128, 8) | 32 | NSA | 6.40  | 3.32  | 1.80  | 1.10  |
> > > > > | (128, 8) | 64 | FSA | 12.00  | 6.38  | 3.60  | 2.06  |
> > > > > | (128, 8) | 64 | NSA | 13.72  | 7.10  | 4.00  | 2.12  |
> > > > > - On H200 GPUs:
> > > > >
> > > > > | (Block Size, TopK) | Seq Len (K) | Framework | TP=1 (s) | TP=2 (s) | TP=4 (s) | TP=8 (s) |
> > > > > | --- | --- | --- | --- | --- | --- | --- |
> > > > > | (64, 16) | 32 | FSA | 1.95  | 1.12  | 0.70  | 0.46  |
> > > > > | (64, 16) | 32 | NSA | 1.97  | 1.22  | 0.70  | 0.49  |
> > > > > | (64, 16) | 64 | FSA | 4.51  | 2.83  | 1.48  | 1.09  |
> > > > > | (64, 16) | 64 | NSA | 4.61  | 2.81  | 1.51  | 1.16  |
> > > > > | (128, 8) | 32 | FSA | 1.82  | 1.04  | 0.64  | 0.45  |
> > > > > | (128, 8) | 32 | NSA | 2.37  | 1.33  | 0.78  | 0.53  |
> > > > > | (128, 8) | 64 | FSA | 3.61  | 2.13  | 1.34  | 0.98  |
> > > > > | (128, 8) | 64 | NSA | 4.62  | 2.63  | 1.61  | 1.15  |
> > > > >
> > > > > We have incorporated the evaluations on compilation overhead and distributed inference into our paper, as presented in Appendix H and Appendix I.

---

### Official Review · Reviewer_8FpK · 2025-10-31

**Soundness:** 3
**Presentation:** 4
**Contribution:** 4
**Rating:** 8
**Confidence:** 3

**Summary:**

This paper proposes Flash Sparse Attention to enable efficient NSA computation with a varied number of query heads in each GQA group to improve LLM inference.

**Strengths:**

1. This paper aims to address an interesting and important problem in long-context LLM applications.
2. The presentation is good with clear writing.
3. The evaluation results are comprehensive and good.

**Weaknesses:**

1. What precision is used for evaluation? Is it FP8, FP16, or FP32?
2. I am wondering how would the proposed kernel could scale beyond 64K length, especially for inference.
3. I am wondering if the authors could provide any further insights into optimizing the proposed kernel for different GPU architectures, such as Hopper and Blackwell.
4. Some system-related works on sparse attention are missing [1-3].

[1] InfiniGen: Efficient Generative Inference of Large Language Models with Dynamic KV Cache Management, OSDI 2024.

[2] Keyformer: KV Cache Reduction through Key Tokens Selection for Efficient Generative Inference, MLSys 2024.

[3] ALISA: Accelerating Large Language Model Inference via Sparsity-Aware KV Caching, ISCA 2024.

**Questions:**

Please see the weaknesses.

---

> ### Author Response · Authors · 2025-11-20
>
> > W1: What precision is used for evaluation? Is it FP8, FP16, or FP32?
> >
>
> In our evaluation, we used BF16 for training and FP16 for inference. We have revised our paper to clearly specify the precision employed in our experiments.
>
> > W2: I am wondering how would the proposed kernel could scale beyond 64K length, especially for inference.
> >
>
> To address this concern, we conducted additional experiments comparing inference execution latency of FSA, NSA, and Full Attention (FA) on 128K and 256K sequence lengths under various NSA hyperparameter configurations. By fixing the number of generated tokens at 512, the head dimension at 128, the number of query heads at 64, and varying the number of key and value heads, we evaluate scenarios in which a GQA group contains 1 to 8 query heads. These experiments were performed on both H20 and H200 GPUs, and the results are summarized in the tables below.
>
> **Results and discussion:** The experimental results demonstrate that FSA’s performance scales well for extremely long sequences. For inference execution latency: (i) compared to NSA, FSA achieves up to 1.40$\times$ speedup and on average 1.16$\times$ lower kernel latency on H20 GPUs, and up to 1.59$\times$ speedup and on average 1.15$\times$ lower kernel latency on H200 GPUs. (ii) Compared to Full Attention, FSA achieves up to 7.20$\times$ speedup and on average 4.61$\times$ lower kernel latency on H20 GPUs, and up to 4.71$\times$ speedup and on average 2.96$\times$ lower kernel latency on H200 GPUs.
>
> **Tables of experimental results:**
>
> - H20, Block Size = 64, TopK = 16:
>
> | GQA | Seq Len (K) | FSA Inference (s) | NSA Inference (s) | FA Inference (s) | FSA Training (s) | NSA Training (s) | FA Training (s) |
> | --- | --- | --- | --- | --- | --- | --- | --- |
> | 1 | 128 | 1.67  | 2.33  | 4.34  | 2.36  | 4.51  | 12.08 |
> | 1 | 256 | 6.71  | 7.49  | 13.45  | 8.70  | 13.29  | 48.23 |
> | 2 | 128 | 1.12  | 1.42  | 4.32  | 1.79  | 2.71  | 12.04 |
> | 2 | 256 | 4.05  | 4.38  | 13.48  | 6.03  | 8.43  | 48.27 |
> | 4 | 128 | 0.77  | 0.86  | 4.35  | 1.43  | 1.75  | 12.07 |
> | 4 | 256 | 2.72  | 2.75  | 13.47  | 4.66  | 5.98  | 48.24 |
> | 8 | 128 | 0.70  | 0.70  | 4.31  | 1.38  | 1.39  | 12.05 |
> | 8 | 256 | 1.94  | 1.95  | 13.46  | 3.99  | 4.75  | 48.26 |
> - H20, Block Size = 128, TopK = 8:
>
> | GQA | Seq Len (K) | FSA Inference (s) | NSA Inference (s) | FA Inference (s) | FSA Training (s) | NSA Training (s) | FA Training (s) |
> | --- | --- | --- | --- | --- | --- | --- | --- |
> | 1 | 128 | 1.49  | 1.93  | 4.34  | 2.15  | 3.50  | 12.08 |
> | 1 | 256 | 5.65  | 7.81  | 13.45  | 7.56  | 13.65  | 48.23 |
> | 2 | 128 | 1.15  | 1.54  | 4.32  | 1.66  | 2.22  | 12.04 |
> | 2 | 256 | 3.49  | 4.55  | 13.48  | 5.40  | 8.33  | 48.27 |
> | 4 | 128 | 0.70  | 0.74  | 4.35  | 1.35  | 1.52  | 12.07 |
> | 4 | 256 | 2.41  | 2.84  | 13.47  | 4.29  | 5.55  | 48.24 |
> | 8 | 128 | 0.67  | 0.68  | 4.31  | 1.31  | 1.41  | 12.05 |
> | 8 | 256 | 1.87  | 1.99  | 13.46  | 3.74  | 4.17  | 48.28 |
> - H200, Block Size = 64, TopK = 16:
>
> | GQA | Seq Len (K) | FSA Inference (s) | NSA Inference (s) | FA Inference (s) | FSA Training (s) | NSA Training (s) | FA Training (s) |
> | --- | --- | --- | --- | --- | --- | --- | --- |
> | 1 | 128 | 1.06  | 1.29  | 1.88  | 1.12  | 2.17  | 4.70 |
> | 1 | 256 | 4.24  | 4.29  | 4.86  | 4.81  | 6.99  | 18.75 |
> | 2 | 128 | 0.74  | 0.85  | 1.85  | 0.80  | 1.24  | 4.68 |
> | 2 | 256 | 2.46  | 2.49  | 4.88  | 3.04  | 4.12  | 18.77 |
> | 4 | 128 | 0.55  | 0.61  | 1.86  | 0.60  | 0.82  | 4.71 |
> | 4 | 256 | 1.52  | 1.61  | 4.89  | 2.15  | 2.70  | 18.76 |
> | 8 | 128 | 0.46  | 0.46  | 1.84  | 0.50  | 0.52  | 4.67 |
> | 8 | 256 | 1.05  | 1.05  | 4.87  | 1.72  | 2.01  | 18.78 |
> - H200, Block Size = 128, TopK = 8:
>
> | GQA | Seq Len (K) | FSA Inference (s) | NSA Inference (s) | FA Inference (s) | FSA Training (s) | NSA Training (s) | FA Training (s) |
> | --- | --- | --- | --- | --- | --- | --- | --- |
> | 1 | 128 | 0.93  | 1.48  | 1.87  | 0.97  | 2.47  | 4.70 |
> | 1 | 256 | 3.48  | 4.42  | 4.86  | 3.99  | 6.79  | 18.75 |
> | 2 | 128 | 0.66  | 0.94  | 1.85  | 0.70  | 1.40  | 4.68 |
> | 2 | 256 | 2.08  | 2.53  | 4.89  | 2.58  | 3.90  | 18.77 |
> | 4 | 128 | 0.49  | 0.56  | 1.86  | 0.53  | 0.73  | 4.71 |
> | 4 | 256 | 1.38  | 1.56  | 4.87  | 1.87  | 2.44  | 18.76 |
> | 8 | 128 | 0.45  | 0.48  | 1.84  | 0.48  | 0.58  | 4.67 |
> | 8 | 256 | 1.03  | 1.07  | 4.85  | 1.52  | 1.70  | 18.78 |
>
> We have integrated these evaluations in Appendix F of our revised paper.

---

> > ### Author Response · Authors · 2025-11-20
> >
> > > W3: I am wondering if the authors could provide any further insights into optimizing the proposed kernel for different GPU architectures, such as Hopper and Blackwell.
> > >
> >
> > Thanks for this question! We summarize the particular optimizations we implemented in FSA to fit different GPU architectures as below:
> >
> > - To implement a more efficient asynchronous IO-compute pipelining in FSA, we **carefully tune the kernel hyperparameters to fit different GPU architectures** to better overlap memory operations with tensor-core computations. For example, on the Ampere architecture, the feasible configurations for *(pipeline stages, warps)* per thread block are typically limited to (2, 2) and (3, 8). In contrast, Hopper enables a broader set of viable configurations, including (2, 2), (3, 4), and (3, 8). With Hopper’s more advanced asynchronous pipelining capabilities, this expanded configuration space allows for finer-grained tuning and higher performance.
> > - To increase the SM occupancy in FSA, we also **adjust the number of thread blocks** differently across GPU architectures. In particular, we launch more thread blocks on Hopper and Blackwell GPUs than on previous generations. Since Hopper and Blackwell provide substantially more SMs, increasing the number of thread blocks is necessary to ensure higher SM occupancy and better utilization of the hardware.
> > - To reduce memory-access latency on Blackwell, we **tune the number of thread blocks within each thread-block cluster**. Blackwell’s block-clustering mechanism allows thread blocks in the same cluster to communicate *directly* through distributed shared memory (DSMEM), effectively creating a larger shared memory space shared across blocks. By exploiting this feature, we can significantly improve the memory access efficiency and overall performance of FSA.
> >
> > > W4: Some system-related works on sparse attention are missing (InfiniGen, Keyformer, ALISA).
> > >
> >
> > We briefly review these related works and then compare them with FSA.
> >
> > **Related works:** InfiniGen [1], Keyformer [2], and ALISA [3] all focus on optimizing KV-cache management and belong to the class of inference-time sparse attention techniques. *InfiniGen* proposes a prefetching mechanism that dynamically selects the crucial subset of KV tokens for the subsequent transformer layer. This is achieved by performing a minimal rehearsal with the inputs of the current transformer layer and part of the query weight and key cache of the subsequent transformer layer. *Keyformer* introduces a novel score function to effectively identify and retain only the most important key and value tokens. *ALISA* develops a sparse window attention mechanism capable of dynamically preserving globally important tokens (i.e., the previous tokens far from the current token) as well as local tokens (i.e., the previous tokens adjacent to the current token). In addition, *ALISA* presents a scheduling algorithm that decides whether to offload KV caches to the CPU or to recompute portions of the KV cache to reduce memory pressure and latency.
> >
> > **Difference between FSA and related works:**
> >
> > - **From the algorithmic perspective**, FSA preserves the sparse-attention pattern of NSA, making it *natively trainable* — in contrast to prior sparse-attention methods that rely on handcrafted criteria to select important key and value tokens. Instead of using predefined formulas to identify crucial key and value tokens, FSA delegates this selection process to the model itself during pre-training.
> > - **From the system perspective**, the key optimizations introduced in prior related works are orthogonal to FSA’s kernel design. For instance, *ALISA’s* scheduling mechanism for KV-cache offloading targets exclusively the inference decoding phase, whereas FSA’s kernel optimizations accelerate the inference prefill and training phase. Looking forward, these approaches are complementary: decoding phase optimizations from prior works (e.g., *ALISA*) could be integrated into FSA to form a more robust end-to-end sparse-attention solution.
> >
> > Reference:
> >
> > [1] InfiniGen: Efficient Generative Inference of Large Language Models with Dynamic KV Cache Management, OSDI 2024.
> >
> > [2] Keyformer: KV Cache Reduction through Key Tokens Selection for Efficient Generative Inference, MLSys 2024.
> >
> > [3] ALISA: Accelerating Large Language Model Inference via Sparsity-Aware KV Caching, ISCA 2024.

---

> > > ### Comment · Reviewer_8FpK · 2025-11-20
> > >
> > > Thank you for your detailed response. I have no further questions and will maintain my score.

---

> > > > ### Author Response · Authors · 2025-11-21
> > > >
> > > > Thank you for your positive feedback. Your insightful comments have greatly strengthened the quality of our work.

---

### Official Review · Reviewer_AuaY · 2025-11-03

**Soundness:** 3
**Presentation:** 3
**Contribution:** 3
**Rating:** 8
**Confidence:** 3

**Summary:**

The paper proposes Flash Sparse Attention (FSA)—a new kernel implementation for Native Sparse Attention (NSA) that flips the loop order: instead of iterating queries (outer) and KV blocks (inner) as in the vanilla NSA kernel, FSA iterates KV blocks in the outer loop and batches of query tokens in the inner loop. This change targets a practical inefficiency of NSA on modern GPUs when each GQA group has few query heads, which otherwise forces padding and underutilizes tensor cores. FSA adds (i) an indexed, non-contiguous query loader, (ii) a dedicated reduction kernel, and (iii) a separate online-softmax statistics kernel. Experiments on H20/H200 GPUs report up to 3.5× kernel speedups vs. NSA and 1.25× training / 1.36× prefill speedups on Llama3-8B, Qwen3-14B, and Qwen2.5-32B for long contexts.

**Strengths:**

1. The paper is well motivated. It identifies padding-driven inefficiency in NSA for common small-g GQA regimes and addresses it directly via loop reordering.
2. The paper presents a solid kernel design with detailed techniques such as non-contiguous query batching with early termination; decoupled reduction to avoid atomics; precomputed online softmax stats to maintain numerical correctness.
3. The paper provides comprehensive evaluation with microbenchmarks across GPUs and (BK,T) settings, plus end-to-end training and inference on multiple LLMs and sequence lengths (8k–64k).

**Weaknesses:**

1. FSA only provides efficiency gains over NSA when each GQA group has few query heads, which limits its impact.

**Questions:**

typo: L63: "implementation fail" ->  "implementations fail"

---

> ### Author Response · Authors · 2025-11-20
>
> > W1: FSA only provides efficiency gains over NSA when each GQA group has few query heads, which limits its impact.
> >
>
> We acknowledge that FSA currently only offers superior performance over the vanilla NSA kernel when a GQA group contains a small number of query heads. Our evaluations show that FSA typically achieves speedups over vanilla NSA when a GQA group contains fewer than (or equal to) eight query heads.  However, when a GQA group contains more than eight query heads, the performance advantages of FSA diminish, as the vanilla NSA kernel no longer incurs padding-related inefficiencies required to satisfy hardware requirements. On the other hand, we want to emphasize that *FSA is impactful across a wide range of LLMs.* For popular LLMs (See the list below), a GQA group typically contains 1, 2, 4, or 8 query heads — configurations that fall within the regime where FSA delivers significant improvements. On the other hand, only a few LLMs, e.g., Qwen3-235B and Llama3.1-405B, adopt a configuration in which each GQA group contains 16 query heads. We summarize the number of query heads within one GQA group of popular open-sourced LLMs [1-5] in the following table:
>
> | Number of Query Heads Within One GQA Group | Models |
> | --- | --- |
> | 1 | Llama2-7B/13B; Qwen1.5-7B, 14B, 72B; Phi3.5-4B |
> | 2 | Qwen3-0.6B, 1.7B; Gemma2-2B, 9B, 27B; |
> | 4 | Llama3-8B; Qwen3-4B, 8B; Mixtral-8x7B; Phi4-15B |
> | 5 | Qwen2.5-14B, 32B; Qwen3-14B |
> | 6 | Qwen2.5-1.5B |
> | 7 | Qwen2.5-0.5B, 7B |
> | 8 | Llama3.1-70B; Llama3.3-70B; Qwen2.5-3B, 72B; Qwen3-32B |
>
> Because the GQA configurations used in mainstream LLMs fall within the regime where FSA provides performance improvements, we believe that the current FSA kernel design and implementation remain practically valuable.
>
> Reference:
>
> [1] Touvron, Hugo, et al. "Llama 2: Open foundation and fine-tuned chat models." *arXiv preprint arXiv:2307.09288* (2023).
>
> [2] Abdin, Marah, et al. "Phi-4 technical report." *arXiv preprint arXiv:2412.08905* (2024).
>
> [3] Yang, An, et al. "Qwen3 technical report." *arXiv preprint arXiv:2505.09388* (2025).
>
> [4] Dubey, Abhimanyu, et al. "The llama 3 herd of models." *arXiv e-prints* (2024).
>
> [5] Jiang, Albert Q., et al. "Mixtral of experts." *arXiv preprint arXiv:2401.04088* (2024).
>
> > Q1: typo: L63: "implementation fail" -> "implementations fail”.
> >
>
> Thank you for pointing this out. We appreciate the reviewer’s careful reading. We have corrected this typo in the revised version of the paper.

---

### Comment · Area_Chair_nodc · 2025-11-27
**Reviewer-Author Discussion**

Hi Reviewers,

Please kinly and actively participate in the review-author dicussion, raise your further concerns so that the authors can explain more, and make your final decisions.

Best,

---

### Author Response · Authors · 2025-12-03

Dear Area Chair,

We appreciate your time and effort in handling our submission. As the rebuttal phase is coming to an end, we would like to briefly summarize the context to aid you in making the final decision.

> ### **Summary of the Reviews**

All three reviewers provided highly positive feedback on our paper and gave overall ratings of **8, 8, and 6**, acknowledging that our work is well-motivated, addresses a timely and important problem, and is supported by solid technical depth and comprehensive experimental validation.

The reviewers' suggestions primarily centered on requests for some additional experiments and further explanations of implementation details. We summarize these constructive suggestions as follows:

**Demanding additional experiments.** The reviewers requested experimental validations in the following aspects:

- Evaluation at longer sequence lengths (Reviewers *8FpK* and *W7t5*).
- Comparison with FlashDecoding (Reviewer *W7t5*).
- Quantification of GPU memory overhead for buffers (Reviewer *W7t5*).
- Further validation of model accuracy preservation (Reviewer *W7t5*).
- Evaluation of compilation overhead and distributed performance (Reviewer *W7t5*).

**Providing further explanations.** The reviewers suggested providing explanations on the following points:

- General applicability of our work to popular LLMs (Reviewer *AuaY*).
- The precision used for evaluation (Reviewer *8FpK*).
- Insights into optimizing our kernel for different GPU architectures (e.g., Hopper and Blackwell) (Reviewer *8FpK*).
- Related work discussion (e.g., InfiniGen, Keyformer, ALISA) (Reviewer *8FpK*).
- Integration complexity with mainstream frameworks and adaptability to other sparse attention methods (Reviewer *W7t5*).

> ### **Our Responses during Rebuttal and Revisions in the Updated Manuscript**

Our comprehensive responses, supported by new additional experiments (summarized in the rebuttal and Appendices D–I) and expanded explanations, systematically addressed all reviewer concerns. We believe the responses and revisions summarized below fully resolve the issues raised.

**Conducting Additional Experiments:**

- We evaluated FSA, NSA, and Full Attention under very long sequence lengths, i.e., 128K and 256K, showing that FSA consistently achieves significant speedups (added in Appendix F) — Reviewer *8FpK, W7t5*.
- We compared FSA with FlashDecoding, demonstrating up to 5.46× speedup (added in Appendix G) — Reviewer *W7t5*.
- We analyzed intermediate buffer memory overhead, showing that FSA’s additional memory overhead remains manageable (added in Appendix E) — Reviewer *W7t5*.
- We provided further validations demonstrating that FSA preserves model accuracy relative to NSA (added in Appendix D) — Reviewer *W7t5*.
- We benchmarked compilation overhead and distributed performance, showing that FSA has comparable compilation time and superior distributed performance relative to NSA (added in Appendix H, I) — Reviewer *W7t5*.

**Providing Further Explanations:**

- We explained that our method is broadly applicable to popular LLMs — Reviewer *AuaY*.
- We clarified the precision used in our evaluations (updated in Section 4.1) — Reviewer *8FpK*.
- We explained how we optimized our kernel for different GPU architectures — Reviewer *8FpK*.
- We discussed how our approach differs from related work — Reviewer *8FpK*.
- We explained that FSA can be integrated into mainstream training and inference frameworks with minimal engineering effort and can be adapted to certain other sparse attention algorithms — Reviewer *W7t5*.

We sincerely appreciate the constructive feedback from all reviewers and believe our responses and revisions significantly strengthen the paper and comprehensively address the reviewers' core concerns.

Best wishes,

Authors of Submission 8921

---

### Meta-Review · Area_Chair_KA2V · 2026-01-03

**Summary:**

The paper proposes Flash Sparse Attention (FSA), an alternative kernel implementation of Native Sparse Attention that tackles inefficiencies when GQA contain a small number of query heads. Reviewers agree the problem is well motivated, the kernel design is technically solid, and the evaluation is comprehensive across GPUs, sequence lengths, and training and inference settings.

**Reviewer Concerns:**

The initial concerns focused on generality, evaluation at very long sequence lengths, missing comparisons with recent sparse kernels, buffer memory overhead, and limited accuracy analysis. The rebuttal addressed these issues by adding new experiments (e.g., 128K--256K context), comparisons with FlashDecoding, detailed buffer memory measurements, and a broader accuracy evaluation across models and tasks. Remaining limitations (e.g., reduced benefit for large GQA groups) are acknowledged and do not hurt the core contribution.

**Reviewer Scores:**

Reviewer AuaY: Unchanged (8); core concerns addressed.

Reviewer 8FpK: Unchanged (8); explicitly said satisfied by rebuttal and maintained score.

Reviewer W7t5: Likely unchanged (6); major concerns were addressed.

---

### Decision · Program_Chairs · 2026-01-26

Accept (Poster)